# Ultrasound trapping and navigation of microrobots in the mouse brain vasculature

Alexia Del Campo Fonseca[1], Chaim Glück [2,3], Jeanne Droux[3,4], Yann Ferry[1], Carole Frei[1], Susanne Wegener[3,4], Bruno Weber [2,3], Mohamad El Amki [3,4,5] ✉ & Daniel Ahmed [1,5] ✉

The intricate and delicate anatomy of the brain poses significant challenges for the treatment of cerebrovascular and neurodegenerative diseases. Thus, precise local drug delivery in hard-to-reach brain regions remains an urgent medical need. Microrobots offer potential solutions; however, their functionality in the brain remains restricted by limited imaging capabilities and complications within blood vessels, such as high blood flows, osmotic pressures, and cellular responses. Here, we introduce ultrasound-activated microrobots for in vivo navigation in brain vasculature. Our microrobots consist of lipid-shelled microbubbles that autonomously aggregate and propel under ultrasound irradiation. We investigate their capacities in vitro within microfluidic-based vasculatures and in vivo within vessels of a living mouse brain. These microrobots self-assemble and execute upstream motion in brain vasculature, achieving velocities up to 1.5 μm/s and moving against blood flows of ~10 mm/s. This work represents a substantial advance towards the therapeutic application of microrobots within the complex brain vasculature.

The human brain, comprising 86 billion neurons, 85 billion non-neuronal cells, and an astounding network of over 650 km of blood vessels, stands as nature's most intricate organ[1]. Despite advancements in biomedical research, numerous brain diseases, including glioblastoma[2,3], Alzheimer's disease[4,5], ischemic stroke[6–8], Parkinson's disease[9,10], schizophrenia[11], epilepsy[12,13], and migraine headache[14] still lack effective therapeutic solutions. In many of these diseases, the main challenge remains the difficulty of accessing the brain[15–19]. However, the emergence of microrobots presents a promising avenue for accessing previously inaccessible regions, including the brain. In particular, these microscopic machines have the potential to navigate the brain vasculature and assist various biological applications such as precise drug delivery and minimally invasive surgery. While biomedical microrobotics has gained increasing attention in recent years[20–27], the application of microrobotics to brain research in vivo remains largely unexplored.

The brain vasculature offers a minimally invasive approach for delivering therapeutic compounds[3,28–35]. However, it also presents significant physical challenges to microrobots, such as adverse flow, the intricacy of the network, and the densely crowded heterogeneous fluidic environment; these impede microrobot progress towards their intended targets. External fields, such as magnetic and ultrasound fields, have demonstrated effectiveness in guiding microrobots within living tissue[36–45]. These manipulations have been achieved in organs with low or no flow such as the stomach[46], urinary bladder[47,48], liver portal vein[49], ear vasculature[50], intestine[51], lung tissue[52], cremaster muscle[53], knee cartilage[54], and cutaneous and subcutaneous vasculature[55,56]. In the context of brain research, previous studies focused on magnetic microrobots showing limited but precise manipulation in in situ mouse models[37]. While magnetic actuation offers precise navigation, its dependence on magnetic particles restricts the biodegradability of microrobots. Ultrasound approaches

[1]Department of Mechanical and Process Engineering, Acoustic Robotics Systems Lab, ETH, Säumerstrasse 4, 8803 Rüschlikon, Switzerland. [2]Institute of Pharmacology and Toxicology, University of Zurich, Winterthurerstrasse 190, 8057 Zürich, Switzerland. [3]Neuroscience Center Zurich, University of Zurich, ETH Zurich, Zurich, Switzerland. [4]Department of Neurology, University Hospital and University of Zurich, and Zurich Neuroscience Center, Zurich 8091, Switzerland. [5]These authors jointly supervised this work: Mohamad El Amki, Daniel Ahmed. ✉e-mail: Mohamad.ElAmki@usz.ch; dahmed@ethz.ch

have yet to demonstrate successful trapping and navigation of microrobots in the brain's physiological settings, but have achieved success in several other contexts. For example, Ghanem et al. demonstrated the acoustic manipulation of glass spheres that, because of their size, were confined primarily to larger cavities such as those found in the urinary bladder[48]. Joss et al. demonstrated the acoustic manipulation of single microbubbles in vessels of zebrafish embryo[57]. Dayton et al. achieved acoustic manipulation of microbubbles within the vasculature of the mouse cremaster muscle, but only in the direction of flow[53]. Finally, Lo et al. succeeded to acoustically trap microbubbles and navigate them inside mouse skin vessels[55].

Here, we introduce ultrasound-activated microrobots that aim to address the current limitations of microscale navigation in brain vessels. Importantly, we focused on the feasibility of actuating microrobots within the brain via ultrasound transmitted through the mouse skull. These microrobots are comprised of gas-filled lipid-coated microbubbles, which undergo self-assembly and become trapped at the walls of blood vessels upon exposure to ultrasound irradiation. While at the walls, the microrobots experience minimal drag from the blood flow, which is conducive to their efficient propulsion. Subsequently, the microrobots translate along the walls, including against blood flow velocities of up to 10 mm/s. To provide real-time in vivo visualization of the microrobots[58–60], we combined the ultrasound navigation capability with two-photon (2P) microscopy (Fig. 1), and we demonstrate the navigation of microrobots in the living mouse brain. Our approach offers a minimally invasive external manipulation method that is robust against high flow rates, reproducible in in vivo environments, and suitable for complex three-dimensional (3D) cerebral capillary networks. These findings contribute to our understanding of ultrasound-based micromanipulation in living brain systems and provide support for the development of novel strategies for targeted drug delivery.

## Results

### Experimental setup

In a first step, we studied the use of ultrasound-controlled microrobots in artificial vessels. We then explored microrobots behavior in vivo in the mouse brain vasculature. To study microrobot manipulation in 3D artificial vessels, we fabricated microfluidic devices made of polydimethylsiloxane (PDMS). PDMS Sylgard was mixed with curing agent in a 10:1 ratio, then carefully poured over 400-μm-diameter copper wires, which served as the template for the vessel structure. After

PDMS curing, the copper wires were carefully removed, resulting in microchannels with circular cross-section. Piezoelectric transducers measuring 3 × 3 mm in size and having a resonance frequency of 490 kHz were positioned on the outer surface of the PDMS channel to generate ultrasound (refer to Fig. 2a). Details regarding the distribution of transducers are provided in the experimental results (see Figs. 2a and 3a and Supplementary Discussion). Each transducer was connected to an electronic function generator to modulate its excitation signal. After fabrication, we injected commercially available fluorescent microbubbles into the microfluidic channels and introduced ultrasound waves via the coupled piezoelectric transducers. The device was mounted on an inverted microscope, and the navigation of microrobots was recorded using high-sensitivity cameras. We further conducted supplementary characterization experiments, encompassing hydrophone pressure measurements, temperature measurements, and acoustic navigation under different scenarios. Detailed explanations of these additional experimental setups can be found in "Methods" and Supplementary Discussion.

We then studied microrobot navigation in vivo in the living mouse brain. A cranial window was surgically implanted, allowing optical access for real-time 2 P microscopy imaging (see Fig. 1 and "Methods"). Fluorescent microbubbles with excitation at a wavelength of 549 nm and emission at 565 nm were subsequently injected into the mouse bloodstream via the tail vein (Supplementary Fig. 1). The in vivo ultrasound navigation setup involved up to four piezoelectric transducers positioned on the sides of the cranial window, arranged orthogonal to each other (Fig. 1a and Supplementary Fig. 2). During in vivo experiments, the transducers were connected to an electronic function generator and an amplifier to modulate their excitation signals, with ultrasound stimulation conducted at frequencies of 490 kHz and voltage ranges of 35–44 $V_{PP}$. The mouse was securely positioned on the stage of the 2P microscope, enabling us to study microbubble dynamics within the brain vasculature.

### Transmission of sound waves from skull to brain

We conducted an investigation on the transmission and attenuation of ultrasound through an ex vivo skull to manipulate microrobots inside the brain tissue. When a piezoelectric transducer is connected to the skull of a mouse, the effective vibrating mass of the system is increased, which affects both the resonant frequency of the transducer and the intensity of the generated sound wave (see Supplementary Fig. 3). For this analysis, we bonded a piezoelectric transducer to an

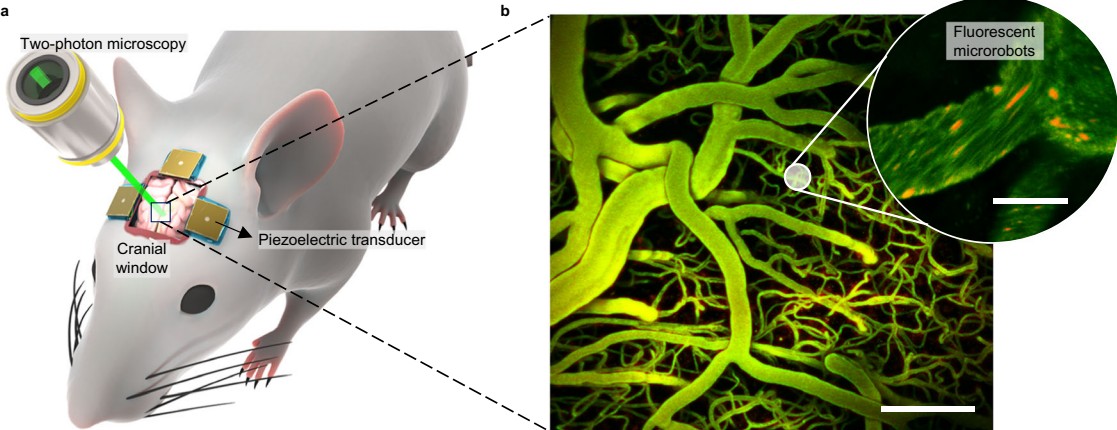

**Fig. 1 | Acoustic microrobot navigation combined with real-time optical imaging. a** Setup for in vivo studies. Optical access to the brain vasculature for 2P microscopy is provided by a cranial window. Piezoelectric transducers are attached to the skull and introduce acoustic waves directed towards the brain. **b** Brain vasculature as visualized by 2P microscopy (z-stack projection). Scale bar is 100 μm. In the inset, fluorescent microbubble-based swarms that will form the microrobots are seen in red. A total sample of 200 microbubble clusters and 39 blood vessels were visualized with this technique during the experiments. For a detailed explanation of the image processing, see "Methods". Scale bar is 10 μm.

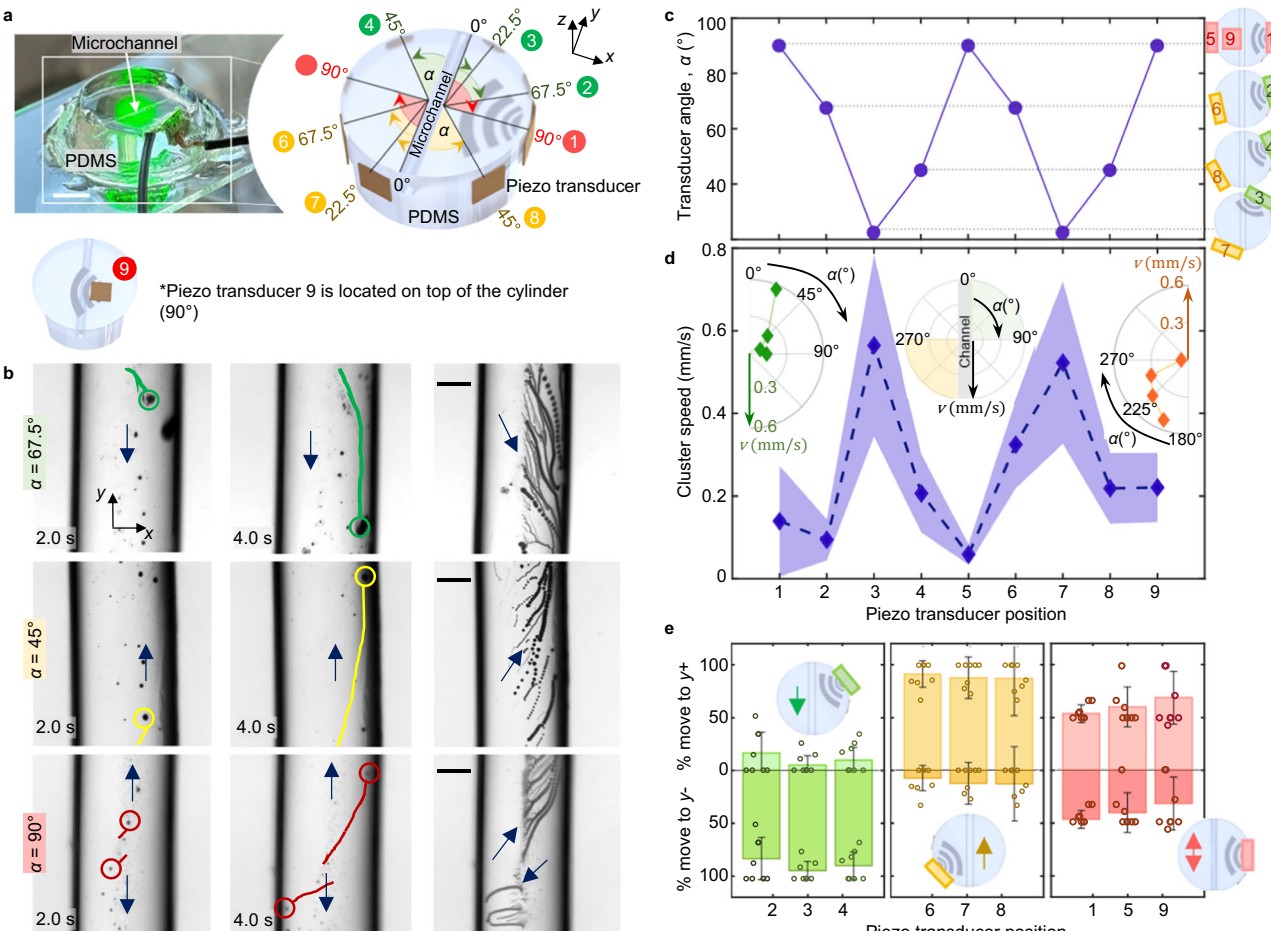

**Fig. 2 | Microrobot navigation in a 3D microfluidic PDMS setup. a** Photo of the experimental setup next to a 3D representation of the PDMS cylinder with transducers. Transducers were coupled to the side of the cylinder and labeled with numbers from 1 to 8. $\alpha$ is the shortest angle between each transducer and the microchannel length. Each transducer has been associated with a color (yellow, red or green), indicating its position along the $y$ axis. Transducer 9 was placed on top of the cylinder, parallel to the microchannel. Scale bar is 0.5 cm. **b** Microscope images of acoustic microrobot navigation under the influence of transducers 3, 8, and 9. Microrobots were manually tracked within the channel, and each track has been colored following the transducer color system. The last column is an image stack spanning four seconds of the navigation process, recorded at eight frames per second. Recordings were done for each transducer activation at least five independent times, all with similar outcomes. The scale bar are 150 μm. **c** Plot of $\alpha$ (shortest angle between transducer and microchannel) for every transducer. The

relative positions of the transducers are illustrated on the side, together with their respective color. **d** Plot of microswarm velocity during navigation with each respective transducer activated. Each point is the average of five measurements, connected to each other by dashed lines and the colored area represents the standard deviation. Inset subplots (in polar coordinates) represent the relation between transducer angle, $\alpha$, and swarm velocity, $v$. Green and orange color correspond to the activation of green and yellow transducers, as defined in (**a**). **e** Plot showing the most probable direction of swarm navigation (up or down the channel) when each transducer is activated. The bar depicts the average of seven independent measurements, which are also shown in the plot as empty circles. Error bars are their standard deviation. Each bar color corresponds to the corresponding transducer color system and thus its position along the $y$ axis; as shown in the inset schematics.

ex vivo mouse skull and measured the transducer's impedance across a frequency range of 100 kHz to 1 MHz with a step size of 1 kHz using an impedance analyzer. The resulting plot is shown in Supplementary Fig. 3a. We observed a slight leftward shift (-15 kHz) in the resonant frequency of the transducer and an attenuation of the vibration intensity (Supplementary Fig. 3a).

Next, we mapped the attenuation of ultrasound pressure within the brain. A 2.5 mm diameter needle hydrophone was positioned below the ex vivo skull inside a deionized and degassed water tank. The piezoelectric transducer was coupled on the skull and positioned opposite to the hydrophone. We measured the resulting pressure field underneath the skull when the transducer was activated at a frequency of 490 kHz and with an amplitude of 45 $V_{PP}$ (see Supplementary Fig. 3b). Maximum pressure was measured when the hydrophone was positioned in close proximity to the transducer, specifically at $y = 0$, refer to Supplementary Fig. 3b. We acquired

pressure data down to a distance of $y = 16$ mm from the skull surface, capturing a consistent decrease in acoustic pressure along the $y$ axis (Supplementary Fig. 3b, c). We further supported these measurements with simulation studies (see Supplementary Fig. 4). To allocate pressure values to specific brain regions, it is important to consider that the mouse skull has an approximate thickness of 2 mm, with protective meningeal layers and cerebrospinal fluid further separating it from the brain. Our investigation demonstrated that the ultrasound pressure directly beneath the skull reached ~100 kPa, while the pressure just below the depth of the pia mater (the innermost meningeal layer) measured around 85 kPa (Supplementary Fig. 3c). These findings highlight the variation in pressure levels at different depths within the cavity of the mouse skull. However, the utilization of real-time tracking of microrobots through 2P imaging introduces additional limitations on the attainable visualization depths. Specifically, 2P microscopy enables visualization at depths of

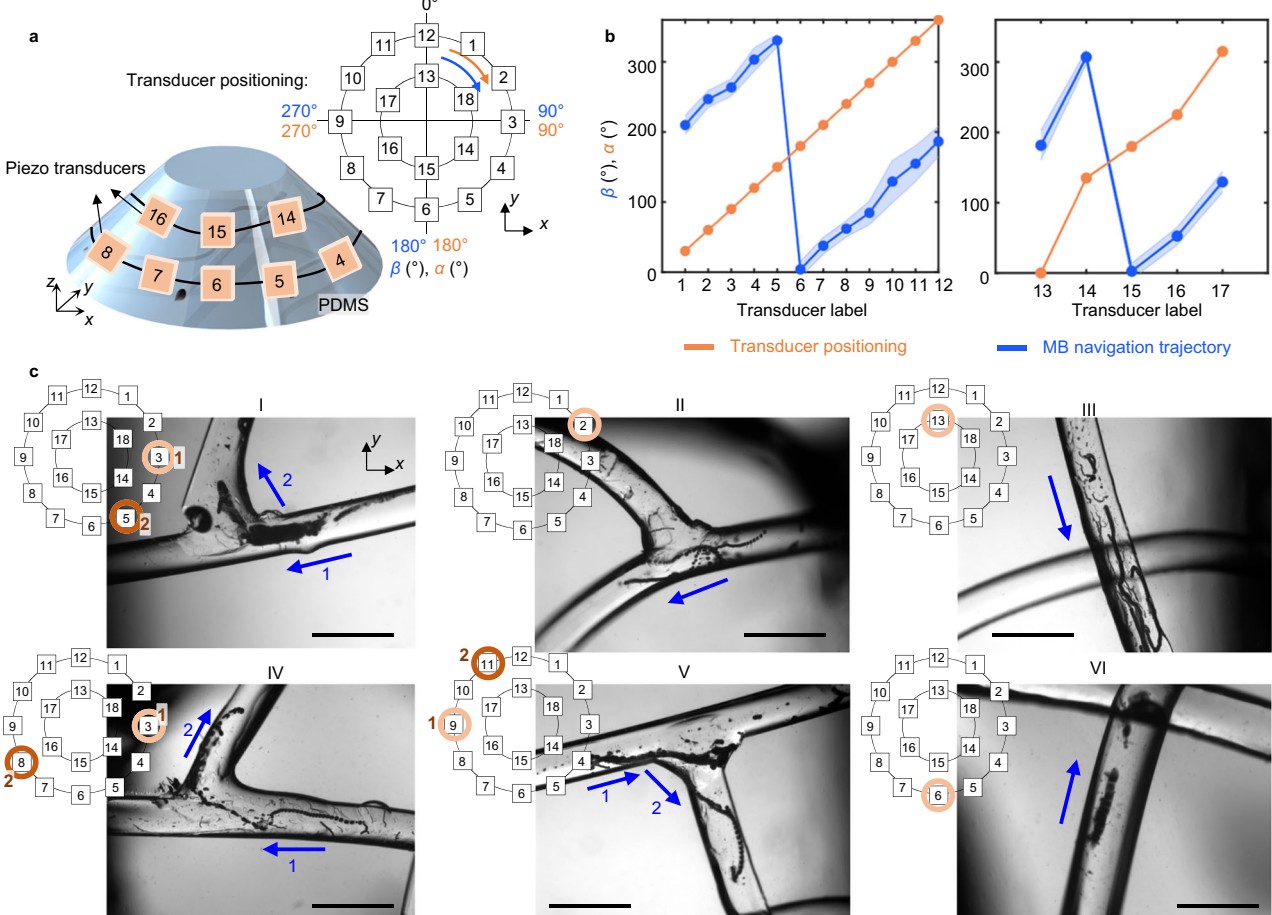

**Fig. 3 | Microrobot navigation in a vascular network using combinatorial transducer actuation. a** Experimental setup consisting of a PDMS in the shape of a cone frustum. Two rows of 18 transducers were coupled to the device. The angles of each transducer ($\alpha$, in orange) and the angle of microrobot directionality ($\beta$, in blue) were indicated as shown in the top right diagram. **b** Plots illustrating the position angle of each transducer ($\alpha$, in orange) and the corresponding direction of microbubble navigation ($\beta$, in blue) for bottom and top row of transducers, respectively. Each blue point represents the mean $\beta$ computed from at least five independent microrobots. The blue error bands show the standard deviation of all measurements from the mean value. **c** Microscope images of the PDMS vasculature network during navigation experiments via combinatorial transducer activation. Each image includes a schematic inset, representing the transducer that was activated to achieve the intended navigation trajectory. In the case of combining two transducers, we have marked as 1 (light brown circle) the transducer that was first activated and 2 (dark brown circle) the transducer that was second activated. Inside the microscope image, we have marked with a blue arrow the trajectory of microswarms upon each transducer activation. We conducted six different experiments, and each image is the result of a stack of frames recorded throughout the experiment. Each experiment was repeated at least 3 times to ensure reproducibility. Scale bar is 1 mm.

up to 500–600 µm in the mouse brain. Within that range, the measured ultrasound pressures ranged from 70 to 80 kPa. Since more than 70% of the ultrasound energy is transmitted to these tissue depths, our setup remains feasible for ultrasound navigation and simultaneous visualization of microrobots through the in vivo skull (see Supplementary Fig. 5).

### Microfluidic validation of acoustic 3D navigation technique

To validate the effectiveness of our ultrasound 3D microrobot navigation technique in in vivo vasculature networks, we conducted preliminary tests in in vitro settings. Our focus was on investigating the self-assembly and navigation capabilities of the microrobots within 3D arbitrary microfluidic channels. The microrobots were created through the ultrasound-initiated self-assembly of individual gas-filled microbubbles, which is achieved through secondary Bjerknes forces; the resultant swarms are then propelled along vessels by primary radiation forces[61]. While our previous work demonstrated motion control limited to 2D movement, in this study, we leveraged the combination of transducers positioned at different locations to steer the microrobots through 3D vascular systems. We have already established that a transducer facing the microrobots can effectively

translate a swarm of microbubbles in the direction of the wave propagation[61]. In this investigation, we examined how the orientation of the acoustic wave ($\alpha$) and the distance between the transducer and the target vessel affect the movement of the microrobots. By exploring these factors, we aimed to enhance our understanding of microrobot navigation within intricate 3D environments.

We investigated the manipulation and control of microrobots within a straight microchannel of circular cross-section, while systematically varying the incident angle of the ultrasound wave. To accomplish this, we positioned nine piezoelectric transducers, designated as 1–9, along the surfaces of a PDMS cylinder (Fig. 2a). The smallest angle formed between the radius line touching the center of each transducer and the microchannel was denoted as $\alpha$ (22.5°, 45°, 67.5°, and 90°; Fig. 2a, c) and indicated the position of each transducer relative to the microchannel. We also incorporated an $x$–$y$ reference system (refer to Fig. 2a, b) to encode the positions of the piezoelectric transducers in space, with the center of the cylinder's top face designated as the origin ($x$, $y$ = 0,0). Transducers positioned above the origin along the $y$ axis are represented in green. Transducers positioned below the origin along the $y$ axis are represented in yellow. Transducers located on the $y$ = 0 line are represented in red. It is important to

note that these three colors (green, yellow, and red) are consistently used throughout Fig. 2. This arrangement allowed us to study the specific influence of transducer positioning on microrobot movements. During the experiments, each transducer was activated independently at 490 kHz and 15 $V_{PP}$. It was observed that in all cases, at least 80% of microswarms moved away from the transducer (shown in Fig. 2b, e). Specifically, when the green transducers (transducers 2, 3, and 4) were activated, the microswarms propelled along the channel length towards the negative $y$ axis, as represented in Fig. 2b, e. When the yellow transducers (transducers 6, 7, and 8) were activated, the microswarms moved along the channel length towards the positive $y$ axis, see Fig. 2b, e. Finally, red transducers (transducers 1, 5, and 9) induced microswarm formation and translation at equal rates towards the positive and negative $y$ axes, see Fig. 2b, e. Notably, the translation velocity decreased in magnitude as $\alpha$ was increased up to 90°. Figure 2c displays the $\alpha$ for each transducer, while Fig. 2d illustrates the velocity that resulted upon each transducer's activation. These two plots exhibit a mirror image pattern, demonstrating a correlation between transducer angle and microswarm velocity. Subplots in Fig. 2d display the magnitude of microswarm velocity for each angle ($\alpha$) using polar coordinates. All told, our results indicate that a perpendicular channel, for which $\alpha = 0°$ and hence the sound waves are incident at 0°, yields maximum translation velocities of up to 1.2 mm/s. Parallel channels, with $\alpha = 90°$ and hence sound waves incident at 90°, display less efficient microrobot translation with speeds of up to 0.4 mm/s. Thus, transducer positioning determines microswarm movement and impacts microswarm velocity.

We next examined the role of the distance between transducer and the target microchannel in acoustic-induced microswarm translation. We fabricated four PDMS cylinders with a piezoelectric transducer located on their top edge (see Supplementary Fig. 6a). Each device contained microchannels positioned at different distances from the transducer (2.15 mm, 5.72 mm, 10.15 mm and 15.25 mm) (Supplementary Fig. 6). For each channel, we measured microswarm velocity during acoustic actuation at a frequency of 490 kHz and voltage of 2 $V_{PP}$. At 2.15 mm distance from the transducer, microbubbles moved with a velocity of 32.2 ± 20 μm/s. Conversely, at 15.25 mm distance, the microbubbles moved at 2.8 ± 1.1 μm/s. Thus, as the distance between the transducer and the target vessel increased, the wave attenuation became more pronounced (see Supplementary Fig. 6). This phenomenon is attributable to the inverse square law[62], which states that the intensity of a wave diminishes as the distance from the source increases, following the relationship $v \propto 1/r^2$ (1) where $v$ denotes the microbubble's velocity and $r$ the distance to the acoustic source (see derivation in Supplementary Discussion)[62]. It is important to note that acoustic waves are also attenuated by other factors, such as damping when passing through different materials and the presence of inhomogeneities in their path[62]. In the context of in vivo manipulation, we need to consider acoustic attenuation as indicated by the pressure map (Supplementary Fig. 3), where decreasing pressure values indicate decreasing navigation velocities.

## Navigation in vertical, curved, and branched trajectories

Microbubble buoyancy plays a crucial role in the navigation of microbubble swarms within 3D microchannels. These microbubbles contain sulfur hexafluoride gas in their core, allowing them to float in the absence of ultrasounds. To investigate the effects of buoyancy, we fabricated a PDMS device featuring a vertical channel with circular cross-section and 400 μm in diameter. A 45°-glass prism was positioned adjacent to the PDMS device to enable side-view imaging of the microbubbles under a microscope (Supplementary Fig. 7). To see the microbubbles through the glass prism, we used red fluorescent microbubbles shined with green light. In the absence of ultrasound, the microbubbles naturally floated to the top. However, when ultrasound at 490 kHz and 10 $V_{PP}$ was applied, acoustic radiation forces

counteracted the effect of buoyancy, causing the microbubbles to propel toward the bottom of the microchannel with a velocity of 165.1 ± 70.1 μm/s. Our research has demonstrated that acoustic radiation forces not only overcome fluid drag but also buoyancy effects on microbubbles.

We further studied the navigation of microrobots within 3D branched and curved microchannels. To accomplish this, we designed and fabricated a PDMS cone frustum device. This design not only mimics the shape of a mouse head but also provides greater flexibility in positioning the transducer at various angles, thereby introducing more degrees of freedom for manipulation. The PDMS device was fabricated in two different designs, referred to as Design A and Design B. Design A featured an inverted half-spherical chamber with a diameter of 5.5 mm embedded at the center of the frustum (see Supplementary Figs. 8 and 9), while Design B contained a network of interconnected vessels with circular cross-sections measuring 400 μm diameter. In both cases, 18 transducer elements were positioned in two rows on the lateral surface of the device (see Fig. 3a). Thus, acoustic waves were generated from two different heights, namely 0.3 cm and 0.6 cm from the bottom of the PDMS structure. We used two reference systems to specify the locations of the transducers. The first was an $x$−$y$−$z$ system in which the center of the PDMS bottom circumference face was designated as the origin ($x, y, z$ = 0,0,0), see Fig. 3a. Here, the $z$ axis defines the transducer height. The second was a polar coordinate system that described the angle ($\alpha$) formed between the radius line touching the center of each transducer and the line $x$ = 0. All angles were measured in the clockwise direction (see the orange arrow in Fig. 3a).

Design A was employed to characterize the activation of each transducer individually. For each transducer, we measured the angle, $\beta$, between the trajectory line of the excited microrobot and the $x$ = 0 reference line. See blue arrow in Fig. 3a. We observed a consistent 180° phase shift between the directionality angle ($\beta$) and the transducer location angle ($\alpha$), indicating that the microrobots move away from the transducer in the direction of wave propagation (see Fig. 3b).

Design B was utilized to characterize microrobot navigation through a 3D vasculature network, including vessel branching and curved trajectories. For this design, the $x$−$y$ plane served as reference to assess the relative orientation between the embedded microchannels and the 18 piezoelectric transducers, see Fig. 3a, c. We carefully chose the most suitable transducer from the set of 18 available options to navigate microbubbles in each specific vessel, then employed a step-by-step activation sequence of various transducers to guide the microbubbles along a predetermined trajectory. The results are depicted in Fig. 3c, where images I, II, IV, and V show examples of microrobots moving through channel branches. In image I, transducer 3 was initially activated to move the microbubbles from right to left. Subsequently, transducer 3 was deactivated and transducer 5 was activated to move microrobots from bottom to top, towards the second channel. Similar transducer activations are shown for images II, IV and V. Conversely, images III and VI show microrobots proceeding along non-straight trajectories under the effect of a single transducer; even when the channel was not aligned head-on with respect to the transducer, we observed translation along the channel. We also showed the navigation of microbubbles through curved microchannels and long distances, see Supplementary Figs. 10 and 11. All in all, we demonstrated that the combined actuation of transducers at convenient positions can successfully navigate microrobots inside an arbitrary vasculature network (Fig. 3).

## In vivo self-assembly of microrobots in mouse brain vessels

We investigated the formation and stability of microbubble swarms in vivo in the brain blood vessels[40,55]. To monitor microrobot self-assembly in vivo, we used intravital 2P microscopy and imaged the

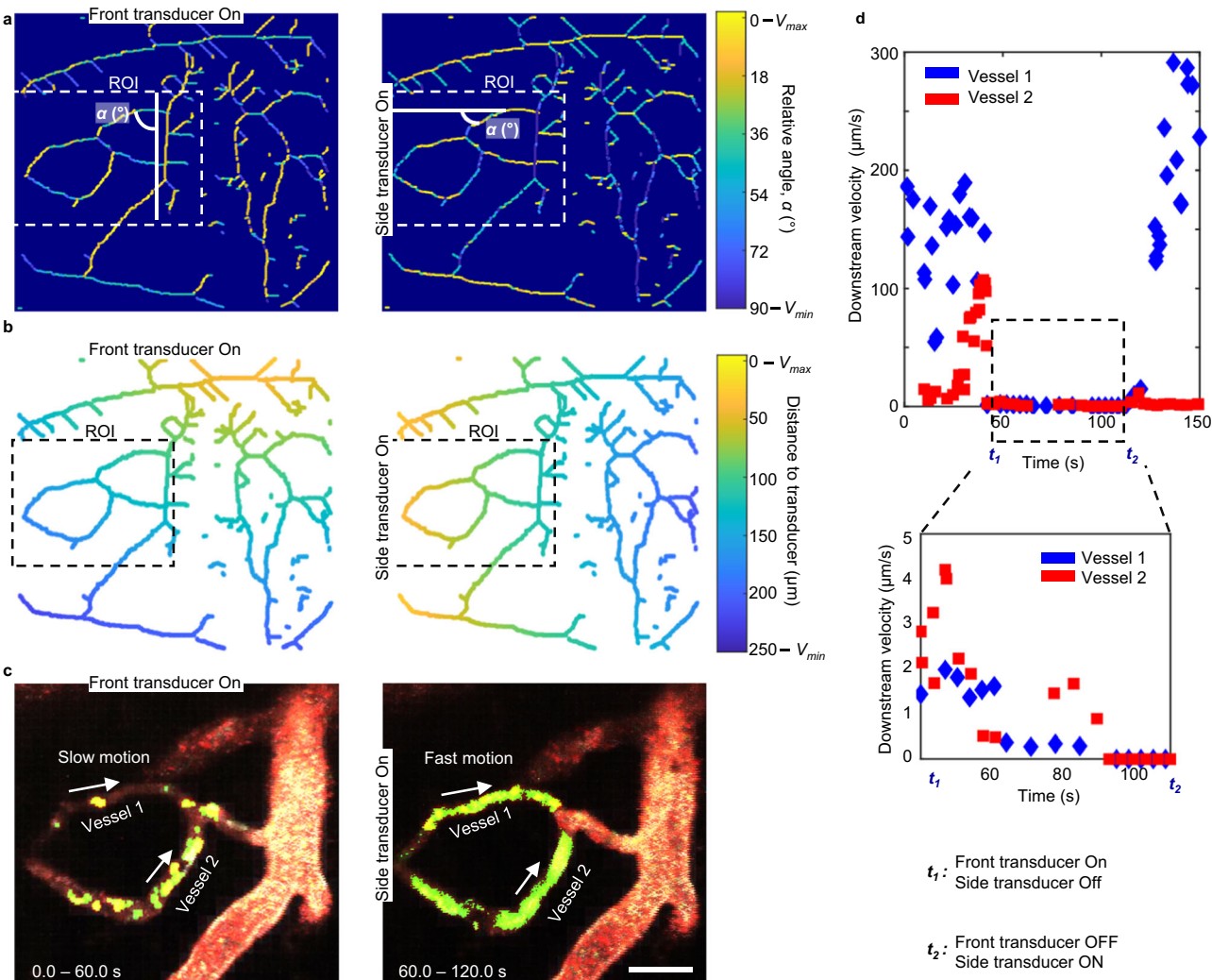

**Fig. 4 | Prediction of microrobot behavior inside a vasculature network and validation in vivo.** First and second columns display front and side transducer activation, respectively. See front and side transducers on the mouse in Supplementary Fig. 16. These positions are also marked along (**a**–**c**). **a** Reconstruction of vasculature lattice via software[63]. Colormap is shown on the right, and it indicates vessels with high ($V_{max}$) or low ($V_{min}$) microswarm navigation velocities due to their relative orientation with the activated transducer, defined by the angle $\alpha$. $\alpha$ is the angle between the line of wave propagation and the vessel length. ROI is the Region of Interest where we performed validation experiments, see (**c**). **b** Reconstruction of vasculature lattice via software. The color assigned to each vessel indicates the distance of the blood vessel to the activated piezo transducer, and thus, the related high ($V_{max}$) or low ($V_{min}$) microbubble velocities. See colormap on the right. ROI is again the Region of Interest. **c** 2P images from the mouse brain vasculature ROI upon activation of front and side transducer, respectively. We have focused on two blood vessels, marked as Vessel 1 and 2. Each image corresponds to a stack of frames recorded during the indicated time. After image processing (see "Methods"), the microswarms are in bright green while single microbubbles are in red. These validations were performed at least five times, within the same specimen. The scale bar is 50 μm. **d** Recorded microswarm downstream, velocities in Vessel 1 (points in blue) and Vessel 2 (points in red). These measurements are extracted from a single acoustic activation experiment, and each point is the velocity of one microswarm at one time point. At time 0, no acoustic signal is being applied. At $t_1$, the front transducer is activated at 45 $V_{PP}$ and 490 kHz. At $t_2$, the front transducer is deactivated, and the side transducer is turned on at 45 $V_{PP}$ and 490 kHz.

brain after intravenous injection of fluorescent microbubbles, see dosage data in Supplementary Fig. 1. After allowing the microbubbles to circulate freely for 5–10 min, one of the acoustic transducers coupled to the mouse skull was activated at 490 kHz and 35 $V_{PP}$. We carefully selected the transducer that was placed more orthogonally to the vessels of interest or target vessels, i.e., ultrasound wave coming at 0°. Our data show that in the presence of ultrasound, microbubbles self-assembled into a swarm and adhered to the vascular wall. During experiments, we targeted pial vessels specifically as those were easy to visualize and access during acoustic actuation. However, it is important to note that there exists an entire network of blood vessels in the background that is also affected by the acoustic signal introduced to the mouse brain. To comprehensively analyze the whole vasculature, we have created a software that predicts the behavior of microbubbles within this intricate framework when activating each individual

transducer, see Supplementary Software[63]. Using 2P microscopy, we captured a ground truth image of the mouse pial vasculature that was located within the cranial window (see Supplementary Fig. 12a, b). We then developed a MATLAB code[63] to store in vectors the relative angles and distances of each vessel with respect to the activated transducers and we reconstructed a color-coded map of the vasculature lattice, see Fig. 4 and Supplementary Fig. 13. The developed software identified vessels that were located orthogonally to the activated transducer (i.e., ultrasound wave coming at 0°) where microbubbles were expected to exhibit more efficient swarm formation and navigation (see Fig. 4a and Supplementary Figs. 12–15). Subsequently, the system generated a visual representation of the direction in which microrobots move upon activating each transducer and the distance of each vessel to the activated transducer, see Fig. 4b. We finally validated the predictive data in vivo in pial vessels (see Fig. 4c, d and Supplementary Figs. 16

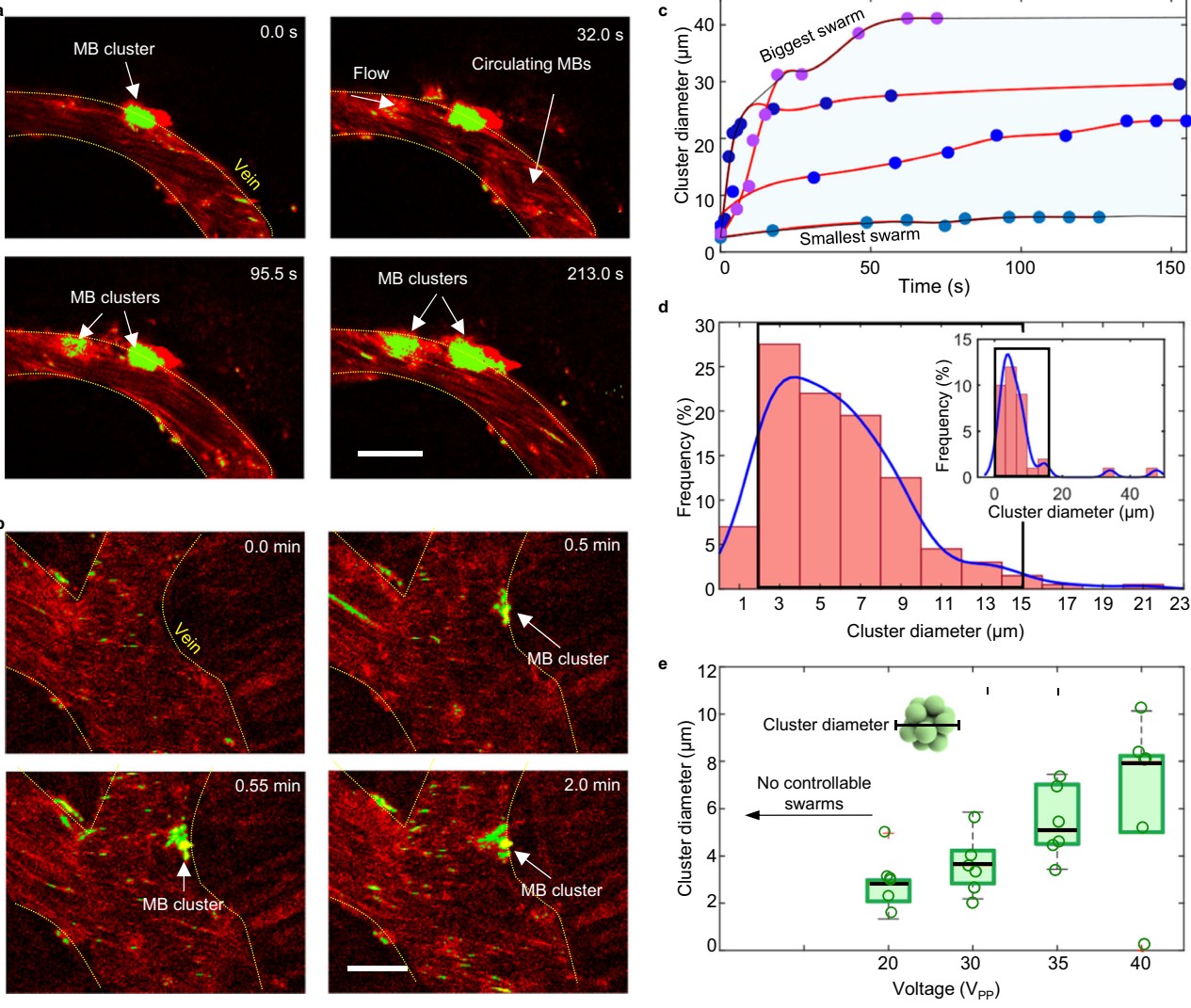

**Fig. 5 | Microbubble-based swarm formation in mouse brain vasculature.**
**a**, **b** Microbubbles (MB) aggregate over time and adhere to the walls of venules under an acoustic signal of 490 kHz and 35 $V_{PP}$. This phenomenon was reproduced in at least ten independent acoustic activation experiments. After image processing (see "Methods"), the microswarms have been colored in bright green and marked with an arrow while non-responding single microbubbles that flow downstream have been colored in red. Scales bar are 50 µm. **c** Evolution of swarm formation and growth under a signal of 490 kHz and 44 $V_{PP}$. Plot shows cluster diameter over time for four different microswarms at four different locations around the brain vasculature. Each microswarm is represented by a different color in the plot. Each point is one microswarm at one time point. Swarm size increases over time until reaching saturation. We colored in light blue the area of possible swarm size evolution, limited by the smallest and biggest swarms observed during experiments.

**d** The inset illustrates a histogram for the size distribution of all the observed microswarms during experiments (from 1 to 40 µm). The majority of swarms range between 3 and 15 µm, which have been marked by a black box. These plots show the cluster diameter distribution observed at an acoustic excitation of 490 kHz and 35 $V_{PP}$. A total sample of 200 clusters was analyzed. Fitting lines in blue have been drawn to summarize the tendency for each plot. **e.** Plot showing cluster diameter versus voltage (at a constant acoustic frequency of 490 kHz). Below voltage 20 $V_{PP}$, we did not observe swarm formation. A schematic inset displays the definition of microswarms diameter. For each voltage value we show six individual measurements (empty circles) and their corresponding boxplot distribution. Black line represents the median and the whiskers are the maximum and minimum values measured during experiments. The cross indicates outliers.

and 17). Specifically, we focused on the activation of the so-called "side transducer", located on the lateral side of the cranial window, and the "front transducer", located on the upper side of the cranial window. We analyzed the movement of microbubbles in two differentiated vessels, labeled as Vessel 1 and 2, see Fig. 4. The resulting microbubble velocities are shown in Fig. 4d. Our results show a strong correlation between the in silico and in vivo data, allowing us to predict precisely microswarm formation around the vascular network. This correlation is also observed for predictions of microrobot directionality and velocity. (Fig. 4d and Supplementary Fig. 14b, c). Altogether, our data show that modeling allowed us to make informed decisions regarding which transducer to activate based on its optimal positioning relative to the whole vasculature lattice.

In vivo, circulating microbubbles were visualized via red fluorescence. In response to transducer activation, we observed that microbubbles started forming aggregations, which we colored green in Fig. 5a, b (see "Methods"). We define successful swarm formation as the formation of microbubble aggregations that are visible under the 2P microscope. We calculated that an average of 3% of the injected bubbles in the brain reacted to ultrasounds by forming swarms (see Supplementary Discussion). These microbubble clusters formed predominantly at vessel walls where they encountered lower drag force from the blood flow (see Fig. 5a, b and Supplementary Movie 1). We also validated the formation of swarms at those regions of the cranial window that were positioned far from the activated transducer (see Supplementary Fig. 18). Importantly, microrobots were able to endure

physiological flow as long as the acoustic signal was activated. Once the transducer was deactivated, the microswarms disassembled and individual bubbles followed the bloodstream.

We then characterized the temporal evolution of swarm size inside blood vessels (Fig. 5c). We observed that as hemodynamic forces brought more microbubbles to the activated site, the clusters grew, resulting in swarms of size up to 40 μm until reaching a saturation state (Fig. 5c). However, not all swarms grew at the same rate. The variability in microswarm formation speed, seen in Fig. 5c, comes from the complexity of the brain vasculature network and the variable relative position of each blood vessel to the activated transducer. In addition, we studied the size distribution of all microbubble swarms that we observed during experiments, see Fig. 5d. We showed that the majority of swarms range was 3–15 μm, with larger swarms found in larger vessels (Fig. 5d). Thus, for the remaining characterization experiments we focused on swarms with these sizes, 3–15 μm. It is still essential to carefully control microbubbles swarms when used in medical applications. During our experiments, we observed incipient cases of vessel clogging in ~3% of the analyzed vessels, primarily in small capillaries and venules. These cases were identified by a sudden reduction in the flow velocity of the affected vessel and the presence of an accumulation of microbubbles (Supplementary Fig. 19). In all these cases, the vascular patency and blood flow were completely recovered when acoustic was turned off.

We also studied the influence of vessel flow velocities on microswarm formation. Here, we observed a higher occurrence of swarm formation at lower flow rates, we show these results in Supplementary Fig. 20. Given that different blood flows are found in different types of vessels, we further investigated the occurrence of swarm formation and navigation in veins, venules, arteries or arterioles (Supplementary Figs. 21 and 22). Hence, we mostly observed microbubble formation in veins and venules (with low flows and diameters of 10–40 μm, refer to Fig. 5d) compared to arteries and arterioles, see Fig. 5d and Supplementary Fig. 21. We also studied the effect of decreasing the applied voltage from 40 to 20 $V_{PP}$; this halving of voltage resulted in the average cluster size also being reduced by half (Fig. 5e). Overall, the multi-parametric observations enabled characterizing the dynamics of microswarm formation in brain vasculature and constitutes a first step towards the implementation of microrobot navigation in vivo.

### Navigation of microrobots inside the mouse brain vessels

Given that high hemodynamic forces within blood vessels create a high challenge for the propulsion and navigation of microrobots, we aimed to demonstrate the navigation capabilities of acoustic microrobots in vivo with and against the blood flow.

We defined successful swarm formation as the formation of visible aggregations of microbubbles that can be observed under the 2P microscope. The formation of microbubble swarms result in an increase in their total volume, amplifying the acoustic radiation forces on them[53]. This phenomenon plays a crucial role in facilitating microrobot navigation. Similarly, successful navigation is defined as the ability of the microbubble swarms to navigate independently of the blood flow stream. Following this definition, we observed microbubble swarm navigation when piezoelectric transducers were activated at 490 kHz and 35 $V_{PP}$. Using 2P imaging, we observed that the process of navigating our microrobots was marked by two behaviors: (1) microswarms continued attracting each other and growing during navigation (Fig. 6a) and (2) the swarms were propelled along the vasculature (Fig. 6b, c and Supplementary Movies 2 and 3), including upstream navigation. We measured microrobots upstream velocities up to 1.5 μm/s, overcoming flow rates up to 10 mm/s (see Fig. 6d). Blood flow was measured at the center of the blood vessels (Supplementary Fig. 23). Compared to our in vitro results, the speed of microrobots within a cerebral vessel was observed to be slower than the speed previously achieved in microfluidic channels, up to 0.8 mm/s. We

propose that these variances can be attributed to the additional acoustic attenuation that occurs within the meningeal layers, brain parenchyma, and blood components, which are not accounted for in the microfluidic models. We also characterized the occurrence of swarm navigation in veins, venules, arteries or arterioles (Supplementary Figs. 21 and 22). Here, we observed that microrobot navigation occurred exclusively in veins and venules (Supplementary Figs. 21 and 22). In addition, we confirmed that microbubble swarm formation and navigation was mostly successful inside vessels with diameters between 10 and 40 μm (see Fig. 6e). Our results showed that microswarm propulsion in the brain vasculature was directly affected by two phenomena. First, microrobot speeds were compromised by internal flow conditions, see Supplementary Fig. 20. Second, the acoustic radiation force on the swarm scales proportionally with its volume, with the higher force experienced by bigger swarms translating to faster navigation[61]. These data explain the formation of large swarms in larger vessels (see results in Fig. 6f) and why microswarms displayed similar upstream velocities independent of swarm size or vessel size (see results in Fig. 6e). Overall, our findings prove the successful acoustic navigation of microrobots upstream against in vivo physiological flows.

Ultimately, we conducted experiments demonstrating the precise control of microswarm movement along designated pathways within the intricate vascular network, encompassing branched vessels within the living brain. As demonstrated in our in vitro experiments, microbubbles could be effectively guided through curved and branched trajectories using combinatorial actuation of transducers. Despite facing challenges such as higher drag forces and dampened acoustic waves, we successfully implemented this technology in vivo. To achieve this, we strategically positioned four transducers, labeled from 1 to 4, around the mouse's head, as illustrated in Fig. 7. As shown in Fig. 7b, we first activated Transducer 3 and the microswarm moved to the left. Subsequently, we activated Transducer 2 (deactivate Transducer 3) and microbubbles moved downwards, towards a second and smaller vessel. By skillfully combining different transducers actuation, we successfully moved microbubbles from one vessel to another, as depicted in Fig. 7b, c and Supplementary Movie 4. These findings highlight the promising capabilities of our approach for navigating microbubbles through complex branched networks in the living brain.

## Discussion

We validated the in vivo feasibility of acoustic microrobots to navigate in the mouse brain vasculature. Prior to conducting in vivo experiments, we conducted preliminary tests in an ex vivo mouse skull and microfluidic channels. During ex vivo experiments, we showed that acoustic damping from the skull does not compromise microrobot navigation within the pial vasculature of the brain (Supplementary Figs. 3–5). In addition, through our experiments in microfluidic device, we showed that the position of an acoustic transducer determines the magnitude and direction of microbubble response. This implies that the transducer positioning impacts the speed and location of swarms within a channel. Nevertheless, in all cases microbubbles move along the vessel wall and in the direction of wave propagation, under the constraints of the boundaries of the channel. Based on our findings, we elucidated that a single acoustic signal generated by a transducer can induce a diverse range of microrobot navigation patterns within a 3D network of microchannels. This is particularly significant in scenarios where the relative positions of the vessels to the transducer are variable, and multiple microbubble swarms are influenced by the acoustic signal. To tackle this situation, our software[63] predicted the vessels within the network that would present higher successful swarm formation and navigation. The predictions from the software were successfully validated during in vivo experiments.

In general, the experiments in vitro demonstrated that microrobots navigation is scalable from single vessels to complex vascular

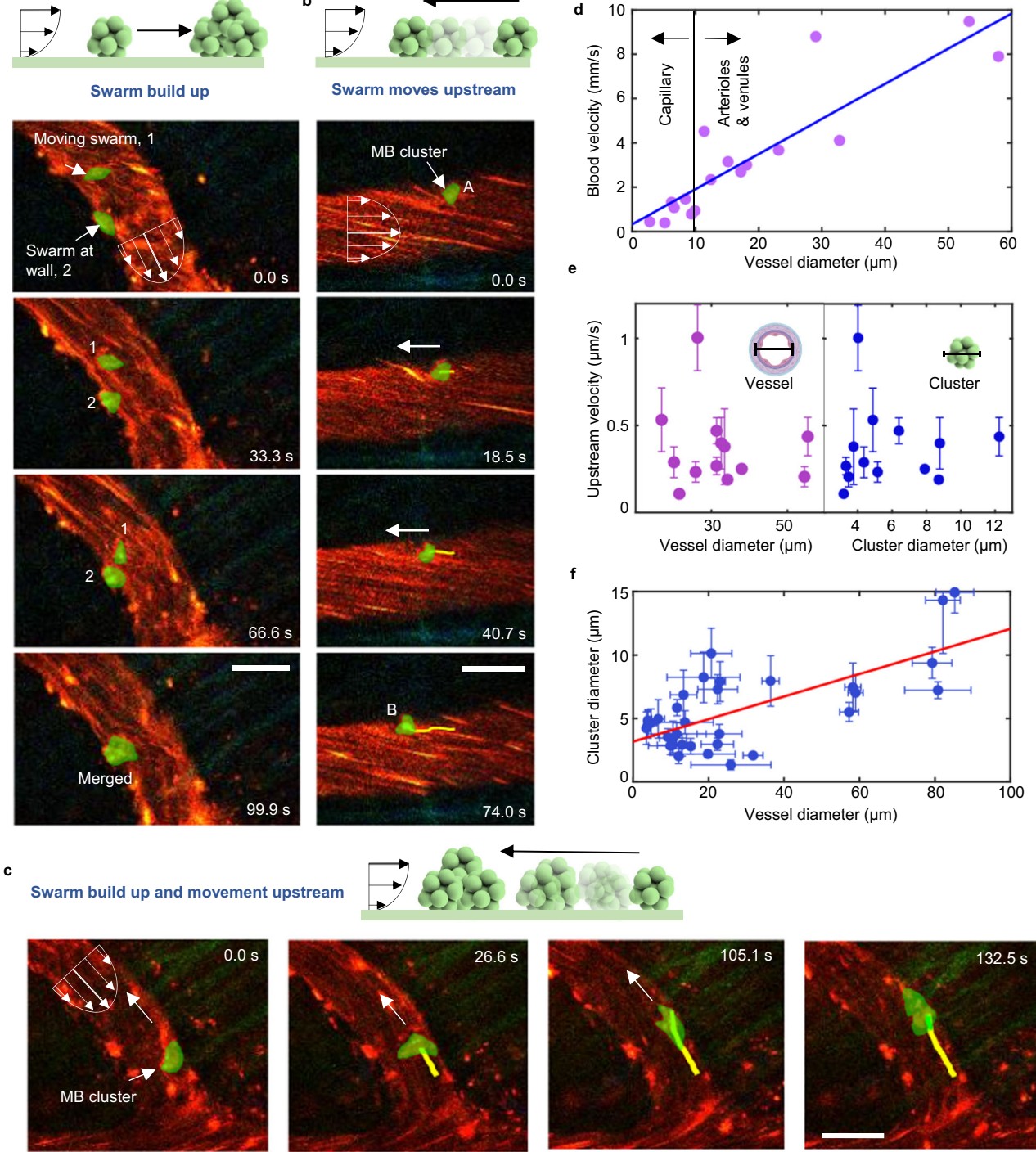

**Fig. 6 | Microswarm upstream navigation in mouse brain vasculature. a** Top schematic illustrates downstream swarm navigation accompanied by simultaneous growth. A microswarm, labeled as 1 (green through manual overlay) is shown inside a venule. Microswarm successfully navigates along a controlled downstream trajectory. Eventually, the microswarm is attracted to a second microswarm labeled as 2. The acoustic signal was 490 kHz and 35 V_PP. Scale bar is 30 μm. This phenomenon was reproduced for at least five independent recordings. **b** Top schematic depicts microswarm upstream movement without growth. A microswarm (indicated by green overlay) is inside a venule. The microswarm is manually tracked (yellow line) as it navigates against the direction of blood flow. The acoustic signal was 490 kHz and 44 V_PP. Scale bar is 30 μm. This phenomenon was reproduced for at least five independent recordings. **c** Top schematic illustrates microswarm navigation upstream accompanied by simultaneous growth. A microswarm (indicated by a green overlay and an arrow) is inside a venule. The microswarm is manually tracked

(yellow line) as it progresses against the flow, gathering more bubbles along its path (Supplementary Movie 3). The acoustic signal was 490 kHz and 35 V_PP. Scale bar is 30 μm. This phenomenon was reproduced in at least five independent acoustic activation experiments. **d** The plot displays the blood flow velocities typically observed in the mouse vasculature. We have drawn a linear fit line in blue with $R^2 = 0.7839$. **e.** Relationship between the observed upstream microswarm velocity versus both the vessel diameter and swarm diameter. Inset schematics represent how both measurements have been taken. The acoustic signal was 490 kHz and 35 V_PP for each measurement. Each point is the average of five independent measurements Data are presented as mean values +/− standard deviation as error bars. **f** Correlation between vessel size and cluster diameter. The acoustic signal for all measurements was 490 kHz and 35 V_PP. Each point is the average of five independent measurements. Data are presented as mean values +/− standard deviation as error bars. $R^2$ for the fitting line in red, is 0.4719.

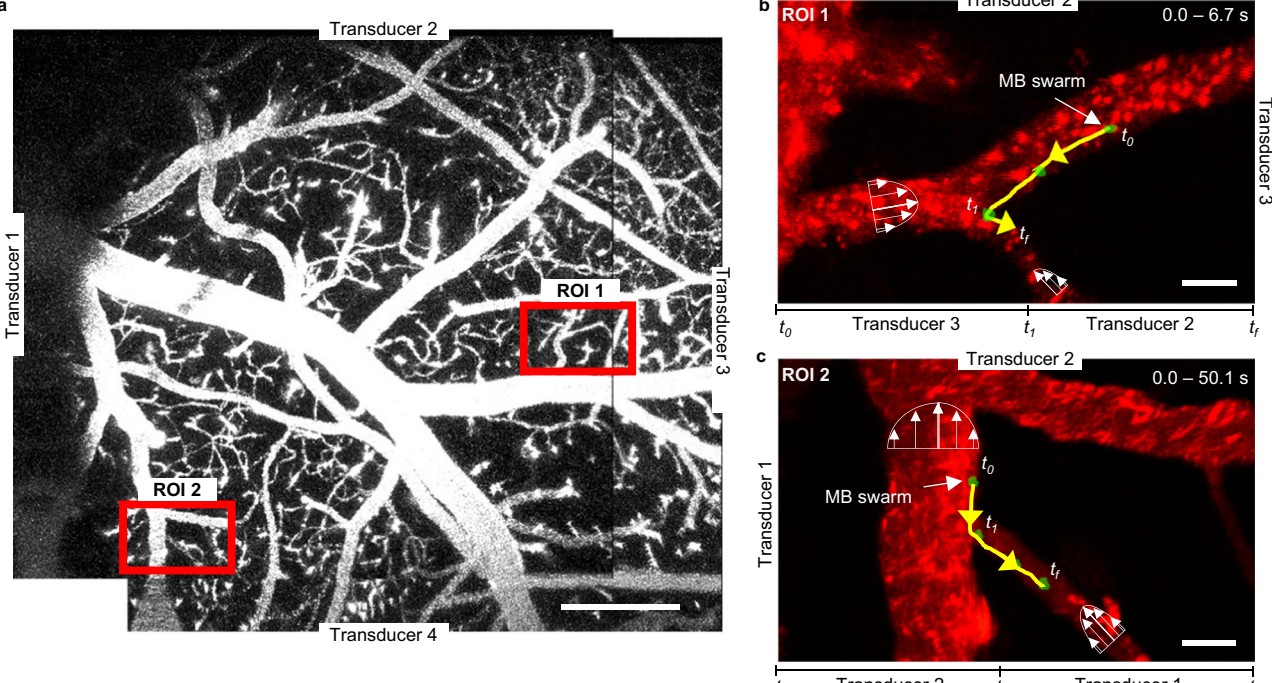

**Fig. 7 | Microswarm navigation through branched vessels in the living mouse brain using combinatorial ultrasound activation. a** Microscope image of the cranial window vasculature network. We used four transducers for microbubble navigation inside this network and we show their location relative to the acquired image. We also show by red squares the spatial locations where we performed microswarm navigation, named as "ROI 1 and ROI 2". Scale bar is 200 μm. **b** Microbubbles are manipulated first left, by piezo transducer 3 and then down by piezo transducer 2. The microswarm is manually tracked (yellow line) as it navigates along a vessel wall, counter to the direction of blood flow. The acoustic signal was 490 kHz and 45 V$_{PP}$. This image results from a stack of 4 different frames taken between 0 and 6.7 s of a single video recording. This behavior was reproduced for three different recordings. Scale bar is 20 μm **c**. Microbubbles are manipulated first down, by piezo transducer 2 and then right by piezo transducer 1. The microswarm is manually tracked (yellow line) as it navigates along a vessel wall, counter to the direction of blood flow. The acoustic signal was 490 kHz and 45 V$_{PP}$. This image results from a stack of four different frames taken between 0 and 50.1 s of video recording. This behavior was reproduced for three different recordings. Scale bar is 20 μm.

networks such as those in the brain. Thus, we translated the results from a microfluidic context to in vivo conditions. For the acoustic implementation, we used a maximum of four transducers, due to the constrained space on the mouse head surface area. While four transducers are adequate to showcase the steering principle for maneuvering microrobots through branches, they do result in reduced precision for achieving efficient wave directionality required for microrobot navigation. Under acoustic excitation, microbubbles aggregated into swarms, and they navigated along the mouse brain vasculature. All the in vivo experiments were performed at a similar tissue depth but at different locations around the head of the mouse, resulting not only in different distances from the transducer, but also different orientations, introducing variability to the acoustic microrobot control (Supplementary Fig. 2). This variability is represented in Fig. 6c, where microrobots are displayed at a range of upstream velocities. Furthermore, we performed a comparison between the vessel sizes where we manipulated microrobots and their average blood flow velocities, showing successful microrobot formation and navigation in flow velocities up to 10 mm/s. Our results reassert the capabilities of acoustic microswarms to work under in vivo physiological conditions. While the initial results in planned navigation of microbubble-based microrobots are promising, there are factors that require further optimization and control. Generating multiple microswarms in the areas affected by acoustic waves is an intrinsic consequence of the technique presented in this study. Eliminating the phenomenon entirely is not currently feasible. Thus, our software-based prediction approach focuses on acknowledging and minimizing its impact to the greatest extent achievable. Our preliminary results show that it is possible to predict microbubble movement along a

defined region of the brain vasculature, as outlined in the Supplementary Discussion; however, we can still not control microswarms in multiple vessels simultaneously. There are currently techniques that tune the shape of the acoustic pressure to specific regions[64,65]. Integrating these techniques with our manipulation technology, particularly in the context of the brain, would yield significant benefits. In addition, current imaging techniques limit real-time non-invasive observation of microscale objects in deeper tissue areas, thus, the manipulation of microswarms in such regions remains a challenge. The actuation system also presents limitations, as seen in Supplementary Movie 3, where multiple microbubble swarms formed but only one of them was able to navigate effectively. While microswarm formation happened more often, navigation was proven to be more difficult to attain (see Supplementary Fig. 22), especially navigation over long distances (see Supplementary Fig. 11).

We use microbubbles with a diameter of approximately 1.1–1.4 μm, and a resonance frequency of 490 kHz. Our working frequency is far from the acoustic resonance of the microbubble, reducing detrimental effects on the surroundings that could arise from behaviors such as microbubble fast strong oscillations or even cavitation. Biosafety and biocompatibility of microbubbles is one important aspect to examine when considering the use of microrobots in vivo, particularly in regard to blood-brain barrier damage, neuronal damage and other potential toxic effects. Immunohistochemical assessment of the brain tissue after ultrasound actuation revealed neither disruption of the vascular endothelium (Supplementary Fig. 24) nor neuronal cell death (Supplementary Fig. 25). Astrocyte activation was minimal and confined to the superficial layer of the cortex (Supplementary Fig. 25). Note that using ultrasound there is a

potential temperature increase that should be considered in further studies and biocompatibility analyses (Supplementary Fig. 26). When targeting deep regions within the brain, various techniques have already been developed to avoid excessive heating in the tissue, i.e., monitoring the temperature, limiting the exposure time, or the use of cooling techniques[66–68]. Nevertheless, our research successfully demonstrated the controlled self-assembly and navigation of microrobots within the living brain vasculature, illustrating their potential as drug carriers for targeted delivery applications.

## Methods

This research complies with all relevant ethical regulations. All animal experiments were approved by the local veterinary authorities in Zurich and conformed to the guidelines of the Swiss Animal Protection Law, Veterinary Office, Canton of Zurich (Act of Animal Protection December 16, 2005 and Animal Protection Ordinance April 23, 2008, animal welfare assurance numbers ZH165/19 and ZH030/23). All experimental measurements are provided as Supplementary Data inside an Excel file named "Source Data".

### In vitro microfluidic setup

We fabricated in vitro channels mimicking the size and mechanical properties of blood vessels via PDMS SYLGARD™ 184 Silicone Elastomer Kit (Sigma-Aldrich). For its fabrication, PDMS SYLGARD™ base was mixed with curing agent in a 10:1 ratio, followed by degassing and pouring into a mold with a wire inside. The wire used was 400 μm in diameter and was previously coated with Silane (Sigma-Aldrich, 440159-100 ml). The PDMS was cured during 1 h at 85 °C. After curing, the wire was pulled out of the PDMS. Piezoelectric transducers were coupled to the sides of the PDMS block, via two-component Epoxy glue (UHU glue, 2 min). A square signal was applied to the piezo elements with a function generator (Tektronix AFG3000) with an amplitude varying from 0.1 $V_{PP}$ to 15 $V_{PP}$. The imaging was realized with an inverted microscope (Zeiss 200 m), and images were recorded with a high-quality camera (Zeiss AxioCAM 305). LUT is linear and covers the full range of the data.

### Microbubbles

Microbubbles used for in vitro and in vivo experiments were commercially purchased from USpheres (Scintica). We used USpheres Tracer-red. Tracer FD-Red uses a dye with excitation at 549 nm and emission at 565 nm. The microbubbles are received inside a glass vial, with a saline solution. We used the UltraMix Agitator (Bio Sonics Trust) for 40 s for microbubble production. The final result is a vial with 0.8 ml and a concentration of ~2.5 × 10¹⁰ bubbles/ml, as detailed by the provider. The size distribution of these bubbles is 1.1–1.4 μm (Supplementary Fig. 1).

### Ex vivo measurement of the intracranial pressure field

The measurement of the acoustic field at the excitation frequency 490 kHz has been done in a deionized and degassed water-filled tank with a 2.5 mm needle hydrophone (HNR, HONDA) coupled to a 20 dB preamplifier (AH-1100 Preamplifier, HONDA). The temperature of the water bath was set to 25 °C. The control computer was driving ten cycles, 67 mV pulse sending from the function generator at a PRF of 100 Hz provided by a digital oscilloscope (PicoScope, 5044D). At the same time, the hydrophones signal output was recorded through the digital oscilloscope for every spatial point in the pre-defined grid. Two varieties of bone were employed for acoustic pressure analysis: a mouse skull obtained from one of the experimental mice, that was sacrificed, and a pig knee bone procured from a local butcher store.

### Piezo transducer characterization

All the piezo transducers used in this document have been commercially purchased through STEMINC and referenced as SMPL3W3T05410. Their dimensions are 3 × 3 × 0.55 mm. An impedance analyzer has been used to measure the impedance of the transducer given an excitation frequency. The measurements consisted of a frequency sweep from 100 kHz to 1 MHz with a 1 kHz step and a 3 ms delay between each measurement. We use 490 kHz as resonance frequency for our experiments.

### Animal preparation and anesthesia

For all experiments, C57BL/6j mice (Charles Rivers, no. 028), three to five months old were used. Both male and female mice were used in the imaging experiment. The mice had free access to water and food and an inverted 12-h light/dark. For the implantation of cranial windows, animals were anesthetized via intraperitoneal injection using a combination of fentanyl (0.05 mg/kg of body weight; brand name: Sintenyl, manufactured by Sintetica), midazolam (5 mg/kg of body weight; brand name: Dormicum, manufactured by Roche), and medetomidine (0.5 mg/kg of body weight; brand name: Domitor, manufactured by Orion Pharma). To ensure proper respiration, a facemask (Narishige International, Japan) was employed, delivering 100% oxygen at a flow rate of 300 ml/min. During 2P imaging, anesthesia was initiated with 4% Isoflurane (Abbott, Cham, Switzerland) in an oxygen/air mixture (100/400 ml/min) and then maintained at 1.2% using a continuous supply of 300 ml/min of 100% oxygen. Throughout all surgical and experimental procedures, the animals' core temperature was maintained at a steady 37 °C using a homeothermic blanket heating system provided by Harvard Apparatus. The animals' heads were securely positioned in a stereotaxic apparatus, and their eyes were kept moistened with vitamin A eye cream (Bausch & Lomb)[69].

### Cranial window surgery

A craniotomy measuring 4 × 4 mm was executed and positioned over the somatosensory cortex. This location was carefully centered ~3 mm away from the Bregma and situated 3.5–4 mm laterally. The procedure was carried out using a dental drill manufactured by Bien-Air. Subsequently, a square coverslip with dimensions of 3 by 3 mm, sourced from UQG Optics, was positioned onto the exposed dura mater. To secure the coverslip to the skull, dental cement (EvoFlow; Ivoclar Vivadent AG) was employed, which was then polymerized using a portable blue light source (600 mW/cm²; Demetron LC).

### Microbubble injection and acoustic activation in vivo

Fluorescent microbubbles (USpheres Tracer-Red, Scintica) and FITC-Dextran (70 kDa, Sigma-Aldrich, cat. No. FD70S) were injected into the bloodstream of the mouse via tail vein injection. In total, 50 μl of the microbubble solution (~2.5 × 10¹⁰ bubbles/ml) was first injected into the blood circulation of the mouse. After 5 min, a second dose of 50 μl was given to the mouse. One single dose of 100 μl was avoided to diminish side effects from the sudden injection of big volumes into the animal. The piezoelectric elements were placed with epoxy two-component glue (UHU, 2 min) to the skull, next to the cranial window (Fig. 1a). Transducers were connected to an oscilloscope (PicoScope, 5044D) for data verification and to a function generator (Tektronix AFG3000). This setup was mounted onto the stage of the 2P microscope, for real-time tracking of microrobots. Two photon images were captured using ScanImage Janelia Research Campus R3.8.1.

### Combinatorial transducer activation implemented on the mouse head

For microbubble navigation in vivo we opted for a simplified setup comprising four transducers (Stemminc SMPL3W3T05410). The transducers we employed had dimensions of 3 × 3 mm, whereas the average width of the mouse head we used was 1.5 cm and length 2 cm. The transducers were linked to a system of switches and interconnected to a single-function generator. This setup enables each transducer to be activated independently, while all of them receive the same acoustic signal.

## Image processing

All the images acquired with an inverted microscope or 2P microscope were open with ImageJ (Fiji). In all of them LUT is linear and covers the full range of the data. No resolution enhancement was utilized in any case. We implemented image processing to the experimental results recorded via 2P microscope during in vivo experiments. Microswarms (bubble aggregations) exhibited high fluorescent intensities; thus, we utilized an intensity-based thresholding on each image to create a mask that only identifies microswarms, excluding single microbubbles within blood flow (Fig. 5). This has been done using ImageJ (Fiji)–Image–Adjust– Threshold and we have employed Otsu thresholding. For the results on Fig. 6, this mask was defined manually, due to poor resolution while the swarm is moving. The swarm mask was set as green and overlaid on the original image, where single microbubbles were tinted in red.

## Biocompatibility analysis—vessel integrity

A mouse C57BL/6 was coupled to two transducers. We applied 20 $V_{PP}$ for three recordings of 200 s each. Then 30 $V_{PP}$ for three recordings of 200 s each. And finally, 40 $V_{PP}$ for six recordings of 200 s each. During the recordings, via 2P microscopy, we injected Hoechst 33342 (Abcam, ab228551) Fluorescent Stain that stains cell nuclei; thus, analyzing vessel integrity. See Supplementary Fig. 24.

## Biocompatibility analysis—histology

After one hour of acoustic activation and microbubble navigation in the mouse, mice were euthanized through an intraperitoneal (i.p.) injection of Pentobarbital (Sigma-Aldrich, Cerilliant®) at an overdose level (200 mg/kg), followed by decapitation. The brains were then extracted and sliced into coronal sections, each 1 mm thick, spanning from 6.5 to 0.5 mm anterior to the inter-aural line. The antibodies that were used for histology were: anti-CD31 (cat. # ab56299, Abcam, 1:100), anti-NeuN (cat. # AB177487, Abcam, 1:1500), and anti-GFAP (cat. # ab4674, Abcam, 1:2000). Tissue sections were incubated with DAPI (4′,6-diamidino-2-phenylindole dihydrochloride) solution (stock 10 mg/ml, 1:10,000, cat. #D9542, Sigma-Aldrich,). See Supplementary Fig. 25.

## Temperature measurements

The recording of the temperature in an ex vivo mice head has been done while three piezo transducers were glued at the same position as in the in vivo experimental setup. A type K thermocouple probe was gently inserted into the brain by the ventral side of the skull. A thermocouple data logger (TC-08, Pico technologies) was used to log and record the thermocouple output on a computer. The three piezo elements were excited with a square signal at the frequency of 490 kHz and with 20 VPP amplitude until the temperature reached equilibrium. See Supplementary Fig. 26.

## Statistics and reproducibility

All the statistical analysis has been executed using MATLAB and MATLAB curve fitting toolbox. All the results presented in the figures depict the average of at least five independent measurements and its associated standard deviation. We have used $R^2$ to assess curve fittings. A total sample of 200 microbubble clusters and 39 blood vessels was studied during the experiments. More details on statistical analysis are found in the corresponding figure captions.

## Reporting summary

Further information on research design is available in the Nature Portfolio Reporting Summary linked to this article.

## Data availability

All data supporting the findings of this study are available within the article and its supplementary files. Any additional requests for information can be directed to, and will be fulfilled by, the corresponding authors. Source data are provided with this paper.

## Code availability

The MATLAB code generated in this study has been deposited in the Zenodo database at https://doi.org/10.5281/zenodo.8279585. License: Creative Commons Attribution 4.0 International. Name of the file: Prediction of microbubble-based microrobot navigation using acoustics inside mouse vasculature. The repository contains three .m files, three readme files with instructions to run the code and raw data for demo running.

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

## Acknowledgements

This project has received funding from the European Research Council
(ERC) under the European Union's Horizon 2020 research and innovation
program grant agreement No 853309 (SONOBOTS) received by D.A.
and ETH Research Grant ETH-08 20–1 received by D.A. The authors thank
Mehmet Fatih Yanik and Mehmet Ozdas for helpful contribution on
acoustic pressure measurements. We thank Mahmoud Medany for
helping us with the electronics.

## Author contributions

D.A. conceived and supervised the project. A.C.F. performed all the
experiments and performed data analysis with contributions from Y.F.
and C.F. and with feedback from M.E.A. and D.A. A.C.F., C.G., Y.F.,
M.E.A., and J.D. performed all the experiments in vivo. A.C.F., C.G.,
M.E.A., and D.A. contributed to the experimental design, scientific pre-
sentation, and discussion. S.W. and B.W. reviewed the manuscript.
A.C.F., C.G., M.E.A., and D.A. wrote and reviewed the manuscript.

## Competing interests

The authors declare no competing interests.

## Additional information

**Supplementary information** The online version contains
supplementary material available at

Mohamad El Amki or Daniel Ahmed.

**Peer review information** *Nature Communications* thanks Li Zhang, and
the other, anonymous, reviewer(s) for their contribution to the peer
review of this work. A peer review file is available.

