## [Peer Review file · Nature Communications]

REVIEWER COMMENTS

Reviewer #1 (Remarks to the Author):

This manuscript studies the acoustic navigation of microrobots in the brain vasculature. The authors investigate the formation and actuation of microbubble swarms in detail and elucidate that the acoustic wave incident angle does not affect the feasibility of microswarm movement. The in vivo experiments in the mice brain are successfully conducted, in which the microrobots are capable of performing upstream motion under flow velocities of up to 10 mm/s. The results in this work may promote the research on applying microrobots in the brain. The main concern is that the authors do not present well in the manuscript whether the proposed microrobot manipulation method has good controllability and broad applicability, which is a key factor for evaluating the significance of this work. It is suggested to provide more experimental results and data to enrich this study and support the proposed points. The detailed comments are as follows.

1. The authors claim that the acoustic microrobots aim to overcome the current limitations of microscale navigation in the brain vessels, but some important fundamental questions have not been well tackled or elucidated. First, microrobots navigated in the brain vessels require good controllability due to the complexity of 3D blood vessels. How to control the movement of microrobots during navigation? For example, when applying in blood vessels with multiple branches, how to navigate the microrobot into the desired branch and/or tortuous vessel (e.g. S-shape) is unclear. Second, although the acoustically driven microrobots move short distances (i.e. about tens of microns) in the blood vessel is demonstrated, continuous driving is required in practical applications. Can longer navigation distances be achieved with this method? And how?
2. Multiple blood vessels exist in the area affected by the acoustic waves, does this mean that multiple microswarms are generated at different locations? If so, these microswarms are likely to be in different states (e.g., moving direction, velocity, and size) due to different relative positions with the transducer. The proposed method seems to can only selectively control one or several microswarms. How to coordinate microswarms across the whole acoustic wave-affected region needs to be considered.
3. The authors used two transducers to actuate microbubble swarms to perform upstream motion in the previous research (ref. 60, DOI: 10.1002/admi.202200877), where one transducer acts to form microswarm and push them towards the side wall, and the other transduce actuates microswarms to move along the wall. In this work, why only a single transducer is needed to achieve this?
4. The complex 3D vascular network of the brain contains blood vessels that extend vertically. Are the acoustically driven microrobots capable of moving against gravity to adapt to 3D blood vessels?
5. What is the dose of fluorescent microbubbles injected from the tail vein in the experiment? How many of them can be successfully trapped and driven by the acoustic wave? What about the biocompatibility of the microbubbles?

6. The application potential of acoustically driven microrobots in the brain may be limited by the penetrating capacity of acoustic waves. Can the acoustic wave penetrate thicker bones, such as human skulls, to actuate the microbubble? Besides, the attenuation of sound waves below the skull is also a concern. Although the authors measured acoustic pressures below an ex vivo skull and claimed that the microrobot control at deeper regions is also attainable (Page 10), this manuscript only presents experimental results at superficial tissue layers (100-200 μm).
7. The microbubble velocities are affected by the relative position and distance to the acoustic transducer (Page 8). However, in Figure 2, only the effect of angle is investigated and not the effect of distance.
8. It says on Page 8 that increased microbubble formation is observed in vessels with low flow compared to vessels with high blood flow velocities. Please give the related data.
9. The microbubble formation and navigation are mostly successful inside veins and capillaries (Page 10). Can you give some detailed results and explain how you assessed the success of formation and navigation?
10. The authors demonstrated successful actuation of microbubbles in the PDMS channel at arbitrary acoustic wave incident angles, which indicates that this acoustic actuation strategy is applicable to complex 3D blood vessels in the brain. However, in the in vivo experiments, it can be seen that multiple microswarms are formed in different parts of the blood vessel within the ultrasound-affected area, why does only one of the microswarms can perform locomotion (Supplementary Video 3)?
11. Under the same acoustic signal (490 kHz and 35 VPP), why does the microswarm in Supplementary Video 1 gather in situ while the microswarm in Supplementary Video 3 moves upstream?
12. Are the acoustic waves that drive microbubbles against flow safe for brain tissue? When need to conduct actuation in deeper regions, the parts close to the transducer are inevitably more affected by the acoustic wave. For example, the measured temperature (Supplementary Figure 5) should be higher in this case.

Reviewer #2 (Remarks to the Author):

The authors describe an imperfect but highly inventive technique to propel and guide clusters or swarms of microbubbles in brain blood vessels. The work is inventive, and the accomplishment is clear, albeit modest, i.e., a few micrometers of transport per second. Nonetheless, creativity carries the day and I support publication. The expanding use of ultrasound will attract others to this approach.

I have several queries:

The authors specify the maximum flow speed that they can move against as 10 mm/s. Is this in the center of the vessel or within a micrometer of the vessel wall?

On page 5 the authors state "... upon skull connection the transducer resonance frequency suffered a subtle shift to the left ...". First, I take it that "shift to the left" means to lower frequency. Second, this sounds like the transducer was mechanically loaded, as a loss mechanism leads to a decrease in the peak frequency of a resonator. What is this loading?

Were the drag forces on the microbubbles the same in pial veins as in pial arterioles?

The size of the microswarm clusters needs to be clarified. Figure 3c shows cluster diameters of up to 30 μm while Figures 3d, 3e, 4e, and 4f imply sizes are less than 15 μm . The text states "... 3 to 10 μm in size." What is it?

The authors state "Microswarms continue attracting each other and grow even during navigation." Is it true that vessels are never clogged by the swarms of microbubbles, even small vessels?

The authors need to state if the vessels in Figures 4a and 4c are venules or arterioles.

In response to Reviewer 1 comments.

This manuscript studies the acoustic navigation of microrobots in the brain vasculature. The authors investigate the formation and actuation of microbubble swarms in detail and elucidate that the acoustic wave incident angle does not affect the feasibility of microswarm movement. The in vivo experiments in the mice brain are successfully conducted, in which the microrobots are capable of performing upstream motion under flow velocities of up to 10 mm/s. The results in this work may promote the research on applying microrobots in the brain. The main concern is that the authors do not present well in the manuscript whether the proposed microrobot manipulation method has good controllability and broad applicability, which is a key factor for evaluating the significance of this work. It is suggested to provide more experimental results and data to enrich this study and support the proposed points.

Response: We greatly appreciate and acknowledge the reviewer's insightful comments and feedback. We agree that the mentioned comments will enrich this study, thus we have performed additional experiments that can clarify how we achieve microrobots control through non-straight trajectories and broader applicability. Per the reviewer's suggestions we have performed experiments on branched structures, curved vessels, vertical vessels, and various distances, to clarify the feasibility of translation to higher complexity models. We also performed further characterization of the navigation approach at different blood flows, vessel types and locations in space; and we have shown additional results on biocompatibility. The new or modified text has been marked in blue within the manuscript and the Supplementary Information. Fragments of this text has also been added to the Reviewer's responses for more clarity.

1. First, microrobots navigated in the brain vessels require good controllability due to the complexity of 3D blood vessels. How to control the movement of microrobots during navigation? For example, when applying in blood vessels with multiple branches, how to navigate the microrobot into the desired branch and/or tortuous vessel (e.g. S-shape) is unclear. Second, although the acoustically driven microrobots move short distances (i.e. about tens of microns) in the blood vessel is demonstrated, continuous driving is required in practical applications.

Response: We thank the reviewer for the comments. We agree with the reviewer that showing microrobot control through 3D complex trajectories will enrich our study. For that, we have performed new experiments to investigate the control of microrobots within complex and three-dimensional vasculature networks resembling those found in the brain. In the former manuscript, a single transducer was used to show microswarm formation and navigation in straight trajectories. Given that microswarms move in the direction of wave propagation, we propose that the activation of multiple transducers at different angles will induce movement of swarms in defined directions. To study that, we have fabricated a 3D arbitrary channel network inside PDMS, and we have exploited the combination of transducers from various orientations, to move microrobots through branched and curved trajectories.

Our new experiments include three dimensional set-ups and complex vasculature designs, including branches (see Fig 3 and Supplementary Fig 9), S-shapes (see Supplementary Fig 10) and vertical vessels (see Question 4 and Supplementary Fig 7); and we show that a combination of transducers helps us to control microrobot directionality in all these cases. We have added five new figures, Fig. 3 and Supplementary Fig 7, 9, 10, 11 to show these results. We have also added these results to the section 'Microfluidic validation of the acoustic 3D navigation technique' (see text in blue, Page 10-11), under the subsection 'Microrobot navigation in vertical, curved and branched trajectories'. We have also included a new section to the Supplementary Information, regarding the manipulation in curved trajectories, as the S-shape (Page S18-19).

New text added to the manuscript (Page 7/10-11):

'We have already established that a transducer facing the microrobots can effectively translate a swarm of microbubbles in the direction of the wave propagation¹. In this investigation, we examined how the orientation of the acoustic wave (α) and the distance between the transducer and the target vessel

affect the movement of the microrobots. By exploring these factors, we aimed to enhance our understanding of microrobot navigation within intricate 3D environments.

.....

We further studied the navigation of microrobots within 3D branched and curved microchannels. To accomplish this, we designed and fabricated a PDMS cone frustum device. This design not only mimics the shape of a mouse head but also provides greater flexibility in positioning the transducer at various angles, thereby introducing more degrees of freedom for manipulation. The PDMS device was fabricated in two different designs, referred to as Design A and Design B. Design A featured a half-spherical chamber with a diameter of 5.5 mm embedded at the centre of the frustum (see Supplementary Fig 9), while Design B contained a network of interconnected vessels with circular cross-sections measuring 400 μm diameter. In both cases, 18 piezo elements were positioned in two rows on the lateral surface of the device (see Fig. 3a). Thus, acoustic waves were generated from two different heights, namely 0.3 cm and 0.6 cm from the bottom of the PDMS structure. We used two reference systems to specify the locations of the transducers. The first was an x-y-z system in which the centre of the PDMS bottom circumference face was designated as the origin ($x,y,z = 0,0,0$), see Fig 3a. Here, the z-axis defines the transducer height. The second was a polar coordinate system that described the angle (α) formed between the radius line touching the center of each transducer and the line $x=0$. All angles were measured in the clockwise direction (see the orange arrow in Fig 3a).

Design A was employed to characterize the activation of each transducer individually. For each transducer, we measured the angle, β , between the trajectory line of the excited microrobot and the $x=0$ reference line. See blue arrow in Fig 3a. We observed a consistent 180° phase shift between the directionality angle (β) and the transducer location angle (α), indicating that the microrobots move away from the transducer in the direction of wave propagation (see Fig 3b).

Design B was utilized to characterize microrobot navigation through a 3D vasculature network, including vessel branching and curved trajectories. For this design, the x-y plane served as reference to assess the relative orientation between the embedded microchannels and the 18 piezoelectric transducers, see Fig 3a, c. We carefully chose the most suitable transducer from the set of 18 available options to navigate microbubbles in each specific vessel, then employed a step-by-step activation sequence of various transducers to guide the microbubbles along a predetermined trajectory. The results are depicted in Fig 3c, where images I, II, IV, and V show examples of microrobots moving through channel branches. In image I, transducer 3 was initially activated to move the microbubbles from right to left. Subsequently, transducer 3 was deactivated and transducer 5 was activated to move microrobots from bottom to top, towards the second channel. Similar transducer activations are shown for images II, IV and V. Conversely, images III and VI show microrobots proceeding along non-straight trajectories under the effect of a single transducer; even when the channel was not aligned head-on with respect to the transducer, we observed translation along the channel. We also showed navigation of microbubbles through curved microchannels, see Supplementary Fig 10. All in all, we demonstrated that the combined actuation of transducers at convenient positions can successfully navigate microrobots inside an arbitrary vasculature network (Fig. 3).

Fig. 3 | Microrobot navigation in a vascular network using combinatorial transducer actuation. **a.** The experimental set-up consisted of a PDMS device in the shape of a cone frustum. Two rows of 18 transducers were coupled to the device. The angles of each transducer (α) and the angle of microrobot directionality (β) were indicated as shown in the top right diagram. **b.** Left: The plot illustrates the position angle of each transducer (orange) and the corresponding direction of microbubble navigation (blue) for the bottom row of transducers. Right: The position angle of the transducers and the direction of microbubble navigation are depicted for the top row of transducers. **c.** We present microscope images of the artificial vasculature network inside the PDMS device captured during microbubble navigation. We conducted six different experiments, and each image is the result of a stack of frames recorded throughout an experiment. Microbubbles were navigated along desired trajectories, including branches and non-straight trajectories. Each image includes a schematic in the top left corner, representing the transducer that was activated to achieve the intended navigation trajectory. Scale bar is 1 mm.

New text added to Supplementary Information (Page S18-19):

Microbubble navigation in curved channels

To further characterize the capabilities of navigation inside various trajectories, we investigated the manipulation of bubbles through curved channels, specifically a channel with an S-shape. We fabricated a PDMS device, in which a wire with an S-shape was embedded. After polymerization, the wire was pulled out and a channel with circular cross-section and 400 μm diameter was engraved in the device. We placed two piezoelectric transducers, as shown in Supplementary Fig. 10, and we activated them each time at 20V_{PP} and 490 kHz. Each transducer faced opposite directions, up or down, and we divided the S-shape channel in two curves, curve 1 was convex and curve 2 was concave. It is important to note that in order to achieve curved trajectories, it is necessary to move in two opposite directions along the trajectory. For instance, when navigating a convex curve, the microbubbles need to move upward until reaching the apex of the curve, and then downward on the other side. As a result, at least a combination of two transducers is required for effective microbubble navigation in curved trajectories.

Upon initial observation, we noticed that the position of the curves and the piezo transducer in our setup allowed for only right-downward movement with Transducer A in the first curve. The acoustic wave from transducer B didn't reach this area with enough intensity, and transducer A was slightly placed to the right, so left side didn't show an organized movement pattern. Conversely, in curve 2, the positioning of the transducers was optimal for bubble manipulation. Transducer A was situated to the left of the curve, while Transducer B was positioned to the right. With Transducer A, bubbles were directed downward until they reached the middle of the curve. Then, Transducer A was deactivated and Transducer B was activated, causing the bubbles to move upward to the other side of the curve. In conclusion, successful navigation through curved channels was achieved, which is particularly relevant as similar geometries are present in in vivo conditions.

Supplementary Fig. 10 | Microrobot navigation in curved vessels. **a.** Experimental set-up. PDMS device with two transducers glued. The set-up was placed on an inverted microscope to facilitate visualization, ensuring that the vessel remains in the same plane and is unaffected by buoyancy when moving through the S shape. **b.** Movement of microbubbles along Curve 1, upon activation of Piezo transducer A at 20 V_{PP} and 490 kHz. The image shown is a stack of 200 frames taken with the microscope during 23 seconds. Scale bar is 400 μm. **c.** Movement of microbubbles along Curve 2, . First activation of Piezo transducer A, at 20 V_{PP} and 490 kHz, moves microbubbles down-right. Then Transducer A is deactivated and Transducer B is activated, so bubbles move top right. The images at the left are a stack images during microbubble navigation. On the right we show the first and last frame of the navigation process, thus first and last position of the microbubbles. Scale bar, 400 μm.

We also recognize the importance of achieving continuous navigation capabilities for microrobots, as highlighted by the reviewer. Hence, we now discuss the navigation of microrobots along long distances. We have performed new experiments to characterize continuous navigation in vitro. For this, we fabricated a straight 3D microfluidic vessel of 13 mm length with a piezoelectric transducer attached to one end of the PDMS device. Our results in microfluidic vessels have shown that the microrobots can navigate continuously along paths up to 10.05 mm, as long as the acoustic wave hits them from an

optimal direction with sufficient intensity. However, in vivo experiments have presented challenges due to the complexity of the brain and the interactions of the acoustic wave with multiple tissues. We have observed that the coupling between the transducer and the mouse skull, as well as scattering of the wave at every interface, can compromise the acoustic wave. Despite these challenges, we agree that continuous driving of microrobots is crucial for in vivo applications. Our experiments have demonstrated that continuous driving is possible using acoustically driven microrobots. It is essential to address the limitations mentioned earlier to achieve successful implementation of this technology in vivo.

We have added a new section in the Supplementary Information (Page S20) and a new Figure (Supplementary Fig. 11) that discusses the results from these new experiments. We also mention now this limitation in vivo within the discussion (Page 16).

Text added to discussion (Page 16):

'The actuation system also presents limitations, as seen in Supplementary video 3, where multiple microbubble swarms formed but only one of them was able to navigate effectively. While microswarm formation happened more often, navigation was proven to be more difficult to attain (see Supplementary Fig. 18), especially navigation over long distances (see Supplementary Fig. 11).'

Text added to Supplementary Information (Page S20):

Manipulation of microrobots for long distances

Our results in microfluidic vessels have shown that the microrobots can navigate continuously along paths up to 10.05 mm, as long as the acoustic wave hits them from an optimal direction with sufficient intensity, see Supplementary Fig 11. However, in vivo experiments have presented challenges due to the complexity of the brain and the interactions of the acoustic wave with multiple tissues. We have observed that the coupling between the transducer and the mouse skull, as well as scattering of the wave at every interface, can compromise the intensity of the acoustic wave.

Supplementary Fig. 11. | Continuous driving of acoustics microrobots. A piezo transducer placed at the top of the image (perpendicular to the channel) was activated at 490 kHz and 10 V_{PP}. We tracked the microrobot down a channel that was 400 μm in cross section diameter and 15 mm long. At the left a scale in mm is shown.

2. Multiple blood vessels exist in the area affected by the acoustic waves, does this mean that multiple microswarms are generated at different locations? If so, these microswarms are likely to be in different states (e.g., moving direction, velocity, and size) due to different relative positions with the transducer. The proposed method seems to can only selectively control one or several microswarms. How to coordinate microswarms across the whole acoustic wave-affected region needs to be considered.

Response: We appreciate and acknowledge the reviewer for the comment. During our experiments we are using a traveling acoustic wave, and as a consequence, a region of the brain is being influenced by the acoustic signal. As outlined by the reviewer, the aforementioned phenomenon elicits the formation of multiple swarms in the surrounding vicinity. The variety of orientations exhibited by the blood vessels in this region results in the presence of diverse swarm responses, such as changes in movement direction, velocity, and size. We before proved that the robots can be activated and controlled *in vivo*; however and per the reviewer's suggestion, we now also discuss the formation of swarms in the background, that happens as consequence of a non-focused ultrasound signal and that results in swarms forming and moving also along the vessels that we don't focus on.

To address this challenge, we have developed a protocol that utilizes imaging data from the brain vasculature network to extract information regarding the distance and orientation of vessels. This information is crucial for planning control over microbubbles in the brain, as it enables us to understand their distribution in the region of interest. Accordingly, in the course of our experimentation, we obtained a baseline image that served as a comprehensive representation of the vasculature network present within the pial region of the mouse brain. Following the acquisition of this image, we employed image processing techniques that enabled the quantification of distance and angle orientation of each vessel with respect to each transducer in the system. Our findings demonstrate that the available data is sufficient to reconstruct the initial skeleton structure of the vasculature network, see Supplementary Fig 12. Additionally, from *in vitro* experiments, we know that when the incident acoustic wave is introduced parallel to the length of the vessel, the microrobots show faster formation and navigation. Thus, with this information we predicted the vessels that were most affected by each transducer in the system (Supplementary Fig. 13). We developed a software that took a skeletonize vasculature image as input and gave as result the color-coded reconstruction of the vasculature lattice, where brighter colors indicate faster microrobot formation and navigation along blood vessels (see Supplementary Fig 13). See a more detailed explanation of this procedure within the main manuscript and the Supplementary Information.

In vivo experiments validated these predictions by showing the formation of swarms in the expected vessels (see Supplementary Fig 14). In the end, these predictions will give us the direction and velocity of bubbles at every location in space, resulting from diverse transducer activations. Although independent control of each of these swarms is still not possible, the predictions enable us to design transducer configuration that moves the targeted microbubbles in a defined direction and that also considers the movement of residual swarms found in the vicinity regions.

Note that still limitations are found when considering deeper regions of the brain. Current imaging techniques make it very difficult to perform real time image of microscale objects inside deeper tissue areas, with minimal invasive approaches. The analysis of swarm formation and navigation at deeper regions was not possible during these studies. Although the prediction could be extended to deeper regions, we couldn't verify its validity. Furthermore, we want to acknowledge that focused ultrasound would reduce the area of influence of the acoustic wave, however the manipulation of microbubbles would be then based on different physical mechanisms.

We have included new text inside the section '***In vivo self-assembly of microrobots in mouse brain vasculature*** (Page 11-12 of manuscript)' to account for this situation, and we have added a detailed explanation in the Supplementary information, including three new supplementary figures (Supplementary Fig 12-14) and the corresponding MATLAB code. See MATLAB code at the end of the Supplementary Information.

Text added to Page 11-12 of main manuscript:

During experiments, we targeted pial vessels specifically as those were easy to visualize and access during acoustic actuation. However, it is important to note that there exists an entire network of blood vessels in the background that is also affected by the acoustic signal introduced to the mouse brain. To comprehensively analyze the whole vasculature, we have created a software that predicts the behavior of microbubbles within this intricate framework when activating each individual transducer, see full code in Supplementary Information. Using 2P microscopy, we captured a ground truth image of the mouse pial vasculature that was located within the cranial window (see Supplementary Fig 12a,b). We then developed a MATLAB code to store in vectors the relative angles and distances of each vessel with respect to the activated transducers and we reconstructed a color-coded map of the vasculature lattice, see Supplementary Fig 13. This enabled to identify vessels that were located orthogonally to the activated transducer (i.e ultrasound wave coming at 0°) where microbubbles were expected to exhibit more efficient swarm formation and navigation (see Supplementary Fig. 12-14). We then validated the predictive data in vivo in pial vessels (Supplementary Fig 14). Specifically, we focused on the activation of the so-called 'side transducer', located on the lateral side of the cranial window. Our results show a strong correlation between the in silico and in vivo data allowing to predict precisely the vessels where microswarm formation and navigation occur (Supplementary Fig 14 b,c). Altogether, our data show that modelling allowed us to make informed decision regarding which transducer to activate based on its optimal positioning relative to the whole vasculature lattice.

Text added to the Supplementary Information (Page S20-S25):

The simultaneous control of multiple swarms formed within the brain region

This study uses a single transducer for the generation of the acoustic signal. The acoustic transducer generates a traveling acoustic wave that penetrates through the tissue and spreads along a defined region within the brain. In our study, we concentrated on a set of particular target vessels to which we directed the navigation of the microrobots. However, it is important to note that microswarms are also being generated concurrently in other vessels located within the acoustic region, and their characteristics are determined by their respective distance and orientation relative to the acoustic transducer.

To address this phenomenon, a comprehensive understanding of the precise location of all the vessels impacted by the acoustic wave is crucial. In order to achieve this, we utilized a two-photon microscope to map the pial vasculature of the mouse, which represents the network of blood vessels located within the superficial region of acoustic actuation. We have processed the image to reduce noise and we have created a skeleton structure of the imaged vasculature; such that each single vessel is represented by a single line that follows the vessel trajectory in space, see Supplementary Fig. 12. Utilizing this skeletal structure, we extracted the orientation of each vessel, which refers to the angle between the vessel and the horizontal axis, as well as the distance between the vessel and the transducer.

We previously demonstrated that the angle and distance of the transducer is a key determinant of the direction in which microbubbles move, see Fig 3. Therefore, by possessing advanced knowledge of the relative position of each vessel with respect to the transducer (i.e., its angle and distance), we are able to anticipate the behaviour of microswarms in each vessel, including their direction of movement and velocity, see Supplementary Fig. 12-14. For this goal, we made the assumption that the piezo transducers were arranged in a circular formation and directed towards the centre. Using this assumption, we determined the relative positioning of each vessel with respect to each transducer in our system. It should be noted that the calculations remain consistent regardless of the number of transducers used. As discussed throughout the manuscript, the optimal placement of the transducer for swarm navigation is perpendicular to the channel, where the line connecting the center of the circumference to the transducer is parallel to the vessel. At this configuration, the swarm velocity is at its maximum, and as the angle deviates from this position, the velocity gradually decreases. Consequently, for each transducer activation, we can identify the vessels that will be most significantly impacted, see Supplementary Fig. 13. On our study we have applied this approach to the vasculature maps of mice and reconstructed the skeleton structure of the vasculature map using a color map, where vessels appearing brighter indicate a higher degree of impact from the actuated piezo. The experiments in vivo proved that the identified brighter vessels showed microswarm formation events, while the darker

vessels didn't, see Supplementary Fig 14. At present, the predictions derived from the vasculature imaging assist us in designing an optimized transducer configuration for the targeted vessels. We anticipate that in the future, an automated processing system capable of analysing the entire vasculature network architecture, combined with machine learning techniques, will permit the simultaneous control of multiple microswarms via dynamic modulation of the acoustic signal. However, there is still challenges to face. Despite significant advances in imaging techniques, limitations still exist when it comes to examining deeper regions of the brain. Real-time imaging of microscale objects inside deeper tissue areas using minimally invasive approaches is incredibly challenging. As a result, we were unable to analyse swarm formation and navigation in deeper regions during our studies, and although the predictions we made can be extended to them, we were unable to verify their validity.

Supplementary Fig. 12. | Two-photon imaging of pial vasculature and extraction of vessel orientation. **a.** Microscopy image of mouse vasculature. The field of view is $460 \mu\text{m} * 460 \mu\text{m}$ square shape. **b.** Post processed image after despeckle, binary transformation and skeleton transformation, via ImageJ software. **c.** Distance histogram, it represents the amount of vessels at each specific distance from the right end of the image. **d.** Combined angle and distance distribution of vessels, it represents the orientation of blood vessels at each specific distance from the transducer. **e.** Image of the vasculature map that has been generated with an original MATLAB code, only using the information from the angles and distances that were extracted from the original image.

Supplementary Fig. 13. | Prediction of swarm velocity and directionality from vasculature map images, example with three transducers. **a.** Microscopy image of mouse brain vasculature. The field of view is $460 \mu\text{m} \times 460 \mu\text{m}$ square shape. **b.** Post processed image after despeckle, binary transformation and skeleton transformation, via ImageJ software. **c.** Swarm velocity map derived for each transducer in the system. We have chosen polar coordinates to illustrate the orientation of blood vessels in space. Red stars mark the position of our transducers, to visualize their relative position with respect to the vessels orientations. Based on the relative orientation between transducer and vessel, we have used a colormap where we find in red the vessels where the swarms will move faster (higher influence of the acoustic wave), and in blue the vessels with lower velocities. **d.** Image of the vasculature map that has been generated with an original MATLAB code using the information from the angles and distances that we extracted from the original image. Specifically here we have focused on the relative orientation between these vessels and the transducer. Brighter colors mean vessels in which higher swarm velocities are found, while darker (more blue) colors represent vessels with low or none microswarm movement.

Supplementary Fig. 14. | Prediction validation during in vivo experiments in pial brain vasculature. **a.** Schematic of transducer positions on the mice. During this experiment we used two transducers that we labeled as front and side transducers. **b.** Image from the two-photon microscope, the relative position of the transducers is shown. In yellow circles we marked the microbubbles that have formed swarms and that are moving due to the activation of the side transducer at 40 VPP and 490 kHz. **c.** Prediction images. This images have been generated with the developed software, and it represents the prediction of the most affected vessels upon activation of side transducer and front transducer respectively.

Text added to the discussion, Page 16:

'While the initial results in planned navigation of microbubble-based microrobots are promising, there are factors that require further optimization and control. Our preliminary results show that it is possible to predict microbubble movement along a defined region of the brain vasculature, as outlined in the Supplementary Info; however, we can still not control microswarms in multiple vessels simultaneously. Additionally, current imaging techniques limit real-time non-invasive observation of microscale objects in deeper tissue areas, thus, the manipulation of microswarms in such regions remains a challenge.'

3. The authors used two transducers to actuate microbubble swarms to perform upstream motion in the previous research (ref. 60, DOI: 10.1002/admi.202200877), where one transducer acts to form microswarm and push them towards the side wall, and the other transducer actuates microswarms to move along the wall. In this work, why only a single transducer is needed to achieve this?

Response: In response to the reviewer's comment, we would like to acknowledge that in our previous paper, we utilized two transducers for two steps of manipulation – a first transducer for assembly and attachment to the wall, and the second for propulsion. Although this was done to illustrate the basic principles of microrobot formation and propulsion, implementing in vivo a system that requires two transducers for each vessel directionality, would be impracticable. Thus, we have now demonstrated that one single transducer also controls microrobot formation, wall adhesion, and propulsion along a vessel length. We have performed new experiments and we have added an explanation to the Supplementary Information (Page S15-17), to better visualize this concept, see Supplementary Fig 8.

Here we want to explain that the role of the first transducer (microrobot self-assembly an attraction to the wall) can also be performed by the second transducer (which originally was only in charge of microrobot propulsion). The aggregation of microbubbles into swarms results from microbubble oscillations when they are inside an acoustic field. The oscillations scatter the sound in all directions, creating a pressure gradient that attracts microbubbles to each other and thus microbubble assembly takes place. It's important to note that this attraction between microbubbles occurs regardless of the direction from which the acoustic field originates. Similarly, the forces responsible for attracting microbubbles to a wall are also unaffected by the incident direction of the acoustic field. In our earlier experiments, we placed a first transducer parallel to the vessel to make sure that microbubbles go straight towards the wall, where they encounter less resistance. Although directing an acoustic wave towards the wall can help microbubbles move directly towards it, it's not mandatory for attracting them to the wall. When oscillating microbubbles are close to the wall, they are naturally pulled towards it by the bubble's interaction with the wall. Based on this information, we can conclude that the acoustic wave generated by the second transducer was sufficient to form microbubble aggregations and to attract the nearby bubbles towards the wall simultaneously propelling them along the length of the blood vessel.

We want to support this explanation with experimental evidence shown in a new Supplementary Fig 8 and 10. We have added this explanation to the Supplementary Information (Page S15-17) 'Microbubble self-assembly and navigation using uniquely one transducer'.

Text added to the Supplementary information (Page S15-17):

Microbubble self-assembly and navigation using uniquely one transducer

In previous studies, the acoustic navigation of microbubble swarms in vitro was shown as a two-step actuation that required two different piezoelectric transducers.² A first transducer located parallel to the artificial vessel, was used to activate microbubble swarm assembly and its movement towards the vessel wall. And a second transducer perpendicular to the vessel was employed to ensure propulsion along its length. Here, we use the same acoustic navigation principle, however we prove that the system can be simplified to a single transducer that achieves all, assembly, adherence to the wall and propulsion. This simplification is necessary for the implementation of this system in vivo, otherwise for each vessel's relative position, we would need two specific transducers to ensure manipulation.

The aggregation of microbubbles into swarms results from microbubble oscillations when they are inside an acoustic field. The oscillations scatter the sound in all directions, creating a pressure gradient that attracts microbubbles to each other and thus microbubble assembly takes place. Importantly, this attraction occurs irrespective of the acoustic field's source, facilitating microbubble assembly regardless of the field's location. Similarly, the secondary Bjerknes forces that drive microbubble attraction to a wall are also independent of the acoustic field's source. Although the presence of a travelling acoustic wave directed towards the vessel wall can aid microbubbles in moving directly towards the wall, this wave is not required for wall attraction. When microbubbles are in close proximity to the wall, they are driven towards it by the wall's attractive force.

Using the same experimental setup as the previous study, we have demonstrated our findings once again, but this time using a single perpendicular transducer (as shown in Supplementary Fig 8). The microfluidic device comprises a channel with a square cross-section of 400 x 30 μm , made of PDMS material, and the piezoelectric transducer is 242 kHz. Here we observe that microbubbles initially self-assemble and move in the direction of wave propagation. The movement of microbubbles is not perfectly straight, and as soon as they get close to the wall, they attach to it and they continue their navigation along the channel wall.

Supplementary Fig. 8. | Formation and navigation of microswarms along a microchannel wall with a single piezoelectric transducer. a. Schematic of the experimental set-up. A single transducer attached to the lateral wall of the PDMS was used for this proof of concept. **b.** Image sequence of microbubble behavior upon acoustic excitation at 242kHz and 3VPP. As seen in the various time frames, microbubbles assemble into a swarm and they move in the direction of wave propagation. The swarms that come close to the PDMS wall, they feel the attraction from the secondary Bjerknes forces

and they adhere to the wall. Once at the wall, swarms keep moving in the direction of wave propagation. Scale bar is 100 μm .

Further experiments in 3D microfluidic set-ups, demonstrated that by activating each time one single transducer, we could achieve formation and navigation of swarms. Note, that in the current system, there are more than one transducer, but each transducer is activated individually each time, with the final goal of steering the microrobot through branches or irregular pathways, see Supplementary Fig. 9. If our goal was only swarm formation and navigation in a straight line, the activation of one transducer would be enough.

Supplementary Fig. 9. | Formation and navigation of microswarms along defined trajectories with a combination of single piezoelectric transducers. **a.** Schematic of the experimental set-up used for the experiment. 8 transducers were attached to a PDMS device. The microfluidic device contained a cross design mimicking an intersection between two vessels. **b.** Image sequence of microbubble behavior upon acoustic excitation at 242kHz and 3VPP of transducers. Transducers were activated in a sequential way, as marked by labels 1 and 2 in the left image. As seen in the various time frames, microbubbles assemble into a swarm and they move in the direction of wave propagation. The swarms can move into the desired branches upon activation of the correct transducers. Scale bar is 500 μm . **c.** Design A. Schematic of the cone frustum PDMS experimental set-up used for experiments. 18 transducers were attached to the PDMS device distributed in two rows. Inside the device we fabricated a half-sphere hollow chamber where we later injected a microbubble containing solution. **d.** Each image consists on a stack of images during acoustic activation. In black we see the trajectory of microbubbles navigation when one transducer is activated. Below each image we have marked in red the activated transducer. Scale bar is 100 μm .

4. The complex 3D vascular network of the brain contains blood vessels that extend vertically. Are the acoustically driven microrobots capable of moving against gravity to adapt to 3D blood vessels?

Response: We thank the reviewer for this comment. Per the reviewer's suggestion we have performed new experiments with vertical channels. During our studies, our focus was on microbubbles that were filled with gas. It is important to note that due to the presence of gas, the gravity forces acting on these microbubbles were naturally counteracted by the buoyancy effect. In the absence of any external forces, these microbubbles naturally rise to the top of vertical channels. We have now studied the movement of acoustic microrobots against the natural buoyancy forces. We added a new discussion regarding buoyancy and vertical channels to the main manuscript (Page 9). Additionally, the results from these experiments are shown in a new Supplementary Fig 7.

We have shown that by properly locating transducers on the top of the vertical PDMS device and using a 45° glass prism to visualize the vertical channel from the microscope, we can move microbubbles against the natural buoyancy force. Microbubbles moved up in the absence of ultrasound signal, while they moved down to the bottom of the channel upon ultrasound activation.

New section to Supplementary Information (Page S14):

Supplementary Fig. 7. | Microbubble navigation in vertical channels. a. Schematic of the experimental set-up, a glass right angle prism was coupled to the face of the PDMS microfluidic device, and it was oriented towards the light of the microscope. On the camera that was mounted to the inverted microscope, we observed a tilted image, where the top side of the PDMS is located at the left of the visualization window (see images in b). b. Microscope time sequence images of microbubbles inside the vertical channel under acoustic actuation of 490 kHz and 10V_{PP}. During Acoustic OFF phase, microbubbles move naturally to the left (to the top of the channel), guided by buoyancy effects. Upon acoustic activation, microbubbles form a swarm, they attach to the side wall and the move to the right (to the bottom of the channel). For these experiments the transducer was placed at the top (so acoustic wave propagated downwards). Scale bar is 100 μm.

Text added to the manuscript (page 9):

'Microbubble buoyancy plays a crucial role in the navigation of microbubble swarms within 3D microchannels. These microbubbles contain sulfur hexafluoride gas in their core, allowing them to float in the absence of ultrasounds. To investigate the effects of buoyancy, we fabricated a PDMS device featuring a vertical channel with circular cross-section and 400 μm in diameter. A 45°-glass prism was positioned adjacent to the PDMS device to enable side-view imaging of the microbubbles under a microscope (Supplementary Figure 7). To see the microbubbles through the glass prism, we used red fluorescent microbubbles shined with green light. In the absence of ultrasound, the microbubbles naturally floated to the top. However, when ultrasound at 490 kHz and 10 V_{PP} was applied, acoustic radiation forces counteracted the effect of buoyancy, causing the microbubbles to propel toward the bottom of the microchannel with a velocity of $165.1 \pm 70.1 \mu\text{m/s}$. Our research has demonstrated that acoustic radiation forces not only overcome fluid drag but also buoyancy effects on microbubbles.'

5. What is the dose of fluorescent microbubbles injected from the tail vein in the experiment? How many of them can be successfully trapped and driven by the acoustic wave? What about the biocompatibility of the microbubbles?

Response: Per the reviewer's suggestions we have added a new section in the Supplementary information to discuss about fluorescent microbubble dosage and the acoustic response of the injected microbubbles (see Page S3-4). First, we explain how the microbubbles doses were administered to the mouse and in what amount. Additionally, by analyzing the videos from our experiments, we have computed the percentage of the total injected bubbles that are being acoustically driven. Given our knowledge of the size of an individual bubble and our assumption that bubble swarms are spherical 3D structures, we were able to estimate the average number of bubbles comprising each swarm by measuring swarm diameter in each experiment. Furthermore, we know microbubble concentration in blood, and the blood flow associated to each vessel, so we could estimate the percentage of bubbles that are in a swarm from the total amount of bubbles that flow through each pial vessel. With this data, we were able to estimate the percentage of bubbles driven by the acoustic wave within the superficial vessels of the brain, from the total number of bubbles initially injected into the mouse. It's important to note that two-photon microscopy do not allow for real time imaging at deeper regions of the brain. Thus, current measurements can only be considered for superficial regions. We couldn't validate these predictions for deeper vessels.

Text in main manuscript (Page 11):

'To monitor microrobot self-assembly *in vivo*, we used intravital 2P microscopy and imaged the brain after intravenous injection of fluorescent microbubbles, see dosage data in Supplementary information (see Fig 4a). After allowing the microbubbles to circulate freely for 5-10 minutes, one of the acoustic transducers coupled to the mouse skull was activated at 490 kHz and 35 V_{PP} .'

Text added to the Supplementary information of the manuscript (Page S3-4):

Fluorescent microbubble dosage and the level of acoustic response

At the beginning of the experiments 50 μl of the microbubble solution ($\sim 2.5 \times 10^{10}$ bubbles/ml) was injected into the blood circulation of the mouse. We visualized the circulation of microbubbles under the two photon microscope. After 5 minutes, a second dose of 50 μl was given to the mouse. These two doses ensured enough microbubble concentration for swarm formation during acoustic activation. One single dose of 100 μl was avoided to avoid side effects from the sudden injection of big volumes into the animal. Note that microbubbles were injected from the tail vein of the mouse and acoustic actuation happens on the brain. It has been shown that the brain of a mouse constitutes 3.5 – 4% of the total blood volume³, thus we can estimate that an order of 10^7 bubbles are circulating inside the mouse brain vasculature.

From our in vivo experiments, we computed the percentage of bubbles in circulation that displayed an acoustic response, which was manifested by their tendency to form swarms. Initially, we determined the number of swarms that had formed within the vessels. Then, we measured swarm diameter and we calculated their volume assuming they had a 3D spherical shape. We approximated the quantity of bubbles forming each swarm by utilizing the diameter of a single bubble, as reported by USpheres, which was 1.1-1.4 μm . In addition, we possessed knowledge regarding the concentration of microbubbles present in the bloodstream, as well as the corresponding blood flow associated with each vessel. Consequently, we were able to calculate the proportion of bubbles that formed a swarm out of the total number of bubbles that flowed through each vessel. Importantly these measurements were taken from experiments in pial vessels of the brain.

Our analysis on the percentage of bubbles exhibiting an acoustic response was found to vary depending on the blood flow of the vessel. These results are discussed later and displayed in Supplementary Figure 16. We also conducted a correlation analysis between vessel types, vessel diameters and the percentage of swarm formation. These results are shown in Supplementary Figure 17. During our experiments we could trap into swarms a maximum of 20% from the circulating bubbles in pial vessel, mostly in capillaries and venules; while bigger vessels like arteries could show values down to 0.5% of bubbles trapped into swarms. In average, we obtained 3% of clustering from the circulating microbubbles in each vessel, during the time of recording. This means an order of 10^6 bubbles will be under the influence of the acoustic signal during the experimental time, if we could excite the whole brain acoustically. It's important to note that two-photon microscopy do not allow for real time imaging at deeper regions of the brain. Thus, current measurements can only be considered for superficial regions. We couldn't validate these predictions for deeper regions.

Additionally, we have included a new section in the Supplementary Information (Page S31-32) regarding biocompatibility. Here we discuss biocompatibility of microbubbles when combined with ultrasound. Note that these microbubbles have been designed specifically for clinical use as contrast agents in ultrasound imaging, their biocompatibility for use in vivo has been previously characterized^{4,5}. Nevertheless, we have performed further analysis on the biocompatibility of bubbles, especially in brain tissue. We injected two consecutive doses of 50 μl of microbubbles to the mouse. After one hour of acoustic activation and microbubble navigation in the mouse, the mouse was sacrificed and perfused for histology studies. Immunohistochemical assessment of the brain tissue after ultrasound actuation revealed neither disruption of the vascular endothelium (Supplemental Figure 21) nor neuronal cell death. Astrocyte activation was minimal and confined to the superficial layer of the cortex. We have added a new Supplementary Figure 21 with these results (Page S31-32), and we refer to this data within the discussion of the main manuscript (Page 16).

Text added to Supplementary Information (Page S31-32)

Biocompatibility studies

Gas filled microbubbles that are surrounded by a lipid layer are often used as contrast agents in medicine, thus numerous studies have already proven their biocompatibility. We additionally assessed the biocompatibility of these bubbles for the brain tissue. We have undergone biocompatibility studies to analyze the effect on brain tissue of oscillating microbubbles.

A C57BL/6 mouse was injected with 50 μL of microbubble solution. After 5 minutes, another 50 μL were injected into the mouse, same dosage as the one used for microbubble navigation experiments. After one hour of acoustic activation and microbubble navigation in the mouse, the mouse was euthanized and perfused for histology studies. We characterized neurons using NeuN staining, astrocytes via the marker GFAP, endothelial cells via the marker CD31 and cell viability using DAPI. Our results show that the acoustic activation of microbubbles does not trigger abnormal neuronal death Endothelial cell lining keeps its integrity; however, astrocyte activation is present at the surface of the tissue, see Supplementary Fig. 21.

Supplementary Fig. 21. | Histology immunostaining for biocompatibility analysis. a. Merged image of a brain slice of the mouse after staining with CD31, DAPI, GFPA and NeuN. Scale bar, 50 μm . **b.** Individual images for each cellular maker, CD31, GFPA, NeuN and DAPI, from left to right, top down. The images show endothelial integrity, astrocyte superficial activation, neuronal survival and tissue integrity. **c.** Merged image of a brain slice of the mouse, at a second differentiated spot, after staining with CD31, DAPI, GFPA and NeuN. Scale bar, 50 μm . **d.** Individual images for each cellular maker, CD31, GFPA, NeuN and DAPI, from left to right, top down. The images show endothelial integrity, astrocyte superficial activation, neuronal survival and tissue integrity.

6. Can the acoustic wave penetrate thicker bones, such as human skulls, to actuate the microbubble? Besides, the attenuation of sound waves below the skull is also a concern. Although the authors measured acoustic pressures below an ex vivo skull and claimed that the microrobot control at deeper regions is also attainable (Page 10), this manuscript only presents experimental results at superficial tissue layers (100-200 μm).

Response: We thank the reviewer for the comment. During our study, we conducted measurements of pressure beneath an ex-vivo mouse skull. However, as noted by the reviewer, it would be beneficial to provide further evidence demonstrating that acoustic waves are capable of penetrating thicker bones and inducing movement of bubbles on the opposite side of the bone. We obtained a pig bone of 5.3mm thickness. Note the values for human skull are in average for men (frontal: 7.8 mm; parietal: 9.6 mm; occipital: 10.1 mm; temporal: 6 mm) and women (frontal: 8.6 mm; parietal: 10.1 mm; occipital: 10 mm; temporal: 6 mm)⁶. We conducted experiments to visualize the behavior of microbubbles within a PDMS material that was coupled to the opposite side of the bone. Our findings demonstrated that acoustic waves are capable of passing through this thickness, and that microbubbles exhibit an acoustic response even at voltages as low as 2 V_{PP}. We have added a new section to the Supplementary Information, discussing the navigation of microbubbles when the acoustic waves pass through thick

bones, and we performed a voltage-based characterization on microbubble velocities at increasing voltages, within this set-up (Page S10-11).

Furthermore, we would like to acknowledge the reviewer's comment regarding the limited depth of our manipulation, as we show manipulation in superficial tissue layers, thus, the pial vasculature of the brain. The reason for this limitation right now is not based on the attenuation of the acoustic wave, but rather on the imaging techniques that can provide with real time imaging of microrobots. Despite significant progress in the field, still imaging at deeper regions within the tissue is difficult to attain; taking into account that we need a non-invasive, real time and high-resolution imaging technique to visualize microrobots navigation. So far, we have provided information on acoustic pressure measurements and experiments that emulate real-world scenarios. However, due to imaging constraints, we have been unable to validate the behavior of microrobots in deeper regions in vivo. We mention this constraint within the main manuscript.

Text added to Supplementary Information (Page S10-11):

We have tested the validity of these piezo transducers for the manipulation of bubbles through thick skulls. For this experiment we obtained a piece of bone from a pig knee with 5.33 mm of thickness. We glued a transducer to one side of the bone and we glued the other side to PDMS. We activated the transducer at 490 kHz and increasing voltage from 2 to 10 V_{PP}. Here, we demonstrated that even at low voltages, the acoustic signal passes through the bone to the microchannel, leading to microswarm formation and navigation. Microbubble velocity was observed to increase with the voltage applied, see Supplementary Fig. 5.

Supplementary Fig. 5. | Actuation of microbubbles through a thick bone. **a.** Experimental set-up. A bone of 5.33 mm thickness was placed in between the piezo transducer and the PDMS with the microchannel. **b.** Image sequence showing microbubbles forming swarms and moving at 490 kHz and 10 V_{PP} of actuation signal. **c.** Voltage characterization. Swarm velocities inside this set-up were analyzed at increasing voltages. The error bar represents the standard deviation and each point represents the average of 5 measurements taken for each voltage value. Scale bar is 100 µm.

7. The microbubble velocities are affected by the relative position and distance to the acoustic transducer (Page 8). However, in Figure 2, only the effect of angle is investigated and not the effect of distance.

Response: We thank the reviewer for the comment. Per the reviewer's suggestion we have now included in the revised manuscript an explanation on how microbubbles are affected not only by acoustic wave incident angle but also by the distance between transducer and the microchannel. We have divided the section 'Microfluidic validation of the acoustic 3D navigation technique' into three subsections: 1. *The impact of incident wave angle on microrobot locomotion*. 2. *The role of distance between transducer and microchannel*. 3. *Microrobot navigation in vertical, curved, and branched trajectories*. Within the study of transducer distance, we show now that when the space between microchannel and transducer is increased, microrobot velocities are reduced. We have proved this experimentally; we have fabricated four PDMS devices with a 3D circular cross section channel at different distances in each device. As distance increases, the attenuation of acoustic pressure increases and acoustic wave intensity decreases following the inverse square law. Thus, the exerted acoustic radiation force on the microbubbles is decreased. We additionally explained how these results need to be adapted to the final acoustic pressure map found in vivo inside the brain.

We have added a reference to the Supplementary Information (Page S11-13), where we explain in detail the effect of acoustic pressure decay following the inverse law and we show the experimental results in Supplementary Figure 6.

New text on the main manuscript (Page 8-9):

The role of distance between transducer and microchannel

We next examined the role of the distance between transducer and the target microchannel in acoustic-induced microswarm translation. We fabricated four PDMS cylinders with a piezoelectric transducer located on their top edge (see Supplementary Fig. 6a). Each device contained microchannels positioned at different distances from the transducer (2.15 mm, 5.72 mm, 10.15 mm and 15.25 mm) (Supplementary Fig. 6). For each channel, we measured microswarm velocity during acoustic actuation at a frequency of 490 kHz and voltage of $2V_{PP}$. At 2.15 mm distance from the transducer, microbubbles moved with a velocity of $32.2 \pm 20 \mu\text{m/s}$. Conversely, at 15.25 mm distance, the microbubbles moved at $2.8 \pm 1.1 \mu\text{m/s}$. Thus, as the distance between the transducer and the target vessel increased, the wave attenuation became more pronounced (see Supplementary Fig. 6). This phenomenon is attributable to the inverse square law⁷, which states that the intensity of a wave diminishes as the distance from the source increases, following the relationship $v \propto 1/r^2$ where v denotes the microbubble's velocity and r the distance to the acoustic source (see derivation in Supplementary Information)⁷. It is important to note that acoustic waves are also attenuated by other factors, such as damping when passing through different materials and the presence of inhomogeneities in their path⁷. In the context of *in vivo* manipulation, we need to consider acoustic attenuation as indicated by the pressure map (Supplementary Fig. 3), where decreasing pressure values indicate in decreasing navigation velocities.

Text added to the Supplementary Information (Page S11-14):

The pressure associated to acoustic waves naturally decays with distance in air, as explained by the inverse square law¹⁰. Here we derived the relation between acoustic radiation force, microbubble velocity, acoustic pressure and distance to the acoustic source, to clarify how microbubble navigation velocities can be affected by the distance of their vessel to the acoustic source. However, note that acoustic waves are also dampened by the material where they are traveling, and by the presence of obstacles and inhomogeneities within their trajectory¹¹. Such that the results obtained from experiments in PDMS demonstrate the tendency of velocity to decay with distance, but the exact decay rate will change when implementing it in vivo.

If we assume that the acoustic pressure follows an inverse proportionality with distance from the source, then we can write:

$P = P_0/r^2$, where p_0 is the pressure at a reference distance and r is the distance to the acoustic source.

The acoustic radiation force, F , acting on a microparticle in an acoustic field is given by:

$F = k\nabla P$, where k is the particle's acoustic contrast factor and ∇p is the gradient of the acoustic pressure. To calculate the gradient of the acoustic pressure, we can take the partial derivatives with respect to the Cartesian coordinates x , y , and z :

$$\frac{\partial P}{\partial x} = -P_0x/r^2, \quad \frac{\partial P}{\partial y} = -P_0y/r^2, \quad \frac{\partial P}{\partial z} = -P_0z/r^2$$

where p_0x , p_0y , and p_0z are the components of the acoustic pressure vector at the reference distance.

The magnitude of the gradient of the acoustic pressure is then:

$$|\nabla P| = \text{sqrt} \left(\left(\frac{\partial p}{\partial x} \right)^2 + \left(\frac{\partial p}{\partial y} \right)^2 + \left(\frac{\partial p}{\partial z} \right)^2 \right) = P_0/r^2$$

Substituting this expression into the relationship between the acoustic radiation force and the gradient of the acoustic pressure, we get: $F \propto P_0/r^2$, which simplifies to: $F \propto 1/r^2$

This shows that the acoustic radiation force decreases with the square of the distance from the source.

The velocity, v , of a microparticle subjected to a constant force, F , can be related to the distance, r , from the acoustic source using Newton's second law: $F = ma = m \left(\frac{dv}{dt} \right)$, where m is the mass of the microparticle and $a = dv/dt$ is its acceleration. Assuming that the force F is constant, we can integrate the above equation to obtain: $v = \left(\frac{F}{m} \right) t + v_0$, where v_0 is the initial velocity of the particle. Substituting the relationship between force and distance, we get: $v = \left(\frac{P_0}{mr^2} \right) t + v_0$. To this decay, we need to add material and inhomogeneities effects. During our experiments, microbubble clusters of size $23.7 \pm 4.3 \mu\text{m}$ decreased their velocity with distance. See Supplementary Fig 6.

Supplementary Fig. 6. | Microbubble navigation velocity at microchannels with different distances from the piezo transducer. **a.** Schematic of the experimental set-up. Four different set-ups were used with channels at four different distances, detailed in the image. The piezoelectric transducer is placed on the top short end of the PDMS device. **b.** Microbubble velocity during swarm navigation versus distance to the transducer. Swarms measured had size of $23.7 \pm 4.3 \mu\text{m}$ to avoid velocity fluctuations due to size. Every point represents the average of five measurements and the error bars represent the standard deviation. In light we have plotted a fitting line that follows a quadratic inverse proportional fit. The logarithmic analysis is shown as a subplot, demonstrating the quadratic fit, $R^2=0.9996$.

8. It says on Page 8 that increased microbubble formation is observed in vessels with low flow compared to vessels with high blood flow velocities. Please give the related data.

Response: We thank the reviewer for the comment. Per the reviewer's suggestion we have added the presentation of this data to the Supplementary information (Page S26-27), and we refer to it inside the main manuscript (Page 13). To show this data, we have plotted the percentage of microbubble clustering seen inside blood vessels, versus the blood flow in each vessel, see Supplementary Fig 16. This plot illustrates, within different blood flows, the proportion of bubbles that form a swarm due to acoustic influence, relative to the total number of flowing bubbles in the vessel. During experiments in vivo the acoustic signal was kept constant, at 490 kHz and 44 V_{PP}, so that we could properly analyze changes on microbubble behaviors at different vessel types and different blood flow velocities. Importantly, we now show the level of clustering formation at various flow rates, where we identified a decreasing tendency for swarm formation, when the blood flow increases.

Text in the main manuscript (Page 13):

'We also studied the influence of vessel flow velocities on microswarm formation. Here, we observed higher occurrence of swarm formation at lower flow rates, we show these results in Supplementary Fig. 16. As mentioned earlier, when flow rates are low, microbubbles experience reduced drag forces, allowing acoustic radiation forces to dominate.'

Text from Supplementary Information (Page S26-27):

To assess the efficacy of microrobot control in vivo, we analyzed how successful was microrobot formation and navigation in blood vessels, and we studied how the vessel intrinsic characteristic (type, size and blood flow) affects the amount of microrobot control. For all in vivo experiments, acoustic signal was kept constant at 490 kHz and 44 V_{PP}.

First, we want to illustrate how higher blood flow values make more difficult microswarm formation. Higher blood flow means higher drag forces that compete with acoustic radiation forces; thus more difficult microbubble aggregations. The calculation of the drag force experienced by a spherical particle in a microscale flow can be performed using the Stokes drag formula, represented by $F_{\text{drag}} = 6\pi\mu aV$. In this equation, F_{drag} denotes the drag force, μ represents the viscosity of the fluid, a stands for the radius of the particle, and V corresponds to the velocity of the fluid relative to the particle. Consequently, blood vessels that are characterized by lower blood velocities, will generate lower drag forces acting upon the manipulated microbubbles. This relationship is illustrated in Fig. 4 of the main manuscript, where the diameter of the blood vessels is demonstrated to be correlated with blood flow velocities.

Consequently, we have calculated the percentage of bubbles forming clusters with respect to the total number of bubbles that flow through each vessel in a period of time. In the section 'Fluorescent microbubble dosage and biocompatibility' within the Supplementary Information, we have explained how we compute the number of bubbles in each formed cluster. During the in vivo recordings, blood flow is fast and manual counting of the amount of flowing bubbles is challenging. However, we know the concentration of bubbles in blood (based on total amount of bubbles injected and the total blood volume of the mouse), and we know the blood flows associated to each vessel diameter, so we can compute the amount of bubbles that flow through a vessel during the recordings time. We have extracted the percentage of bubble clustering and we have plotted it against blood flow rates.

The experimental results show that the percentage of bubbles that form clusters during recordings is lower in vessels with high blood flow, see Supplementary Fig 16.

Supplementary Fig. 16. | Microbubble formation versus blood flow. **a.** Two-photon microscope images from the brain vasculature of the mouse showing microswarm formation over time. The vessel shown is a venule. Microbubbles formed at the vessel walls; and we marked with white circles some examples of swarms that we used for quantification of clustering %. Scale bar, 50 μm . **b.** Plot of the experimental results that show the percentage of clustering at each blood flow present inside blood vessels. Blue area illustrates the decay tendency of clustering % versus blood flow velocities.

9. The microbubble formation and navigation are mostly successful inside veins and capillaries (Page 10). Can you give some detailed results and explain how you assessed the success of formation and navigation?

Response: We thank the reviewer for the comment. We have now clarified how is the assessment of successful formation and navigation of microbubbles, within the main manuscript (page 13).

We defined successful swarm formation as the formation of visible aggregations of microbubbles that can be observed under the two-photon microscope. The formation of microbubble swarms result in an increase in their total volume, amplifying the acoustic radiation forces on them¹⁰. This phenomenon plays a crucial role in facilitating microrobot navigation. Similarly, successful navigation is defined as the ability of the microbubble swarms to navigate independently of the blood flow stream

Additionally, we have performed a more detailed study of the formation and movement of microbubble swarms in different types of vessels, classifying them as venules, veins, arterioles, and arteries. From the same experimental data used previously, we have now computed the number of swarm formation events, the percentage of microbubble clustering and the number of navigation events, at different vessel types. We discuss these results within the Supplementary Information (Page S26-28), and we have included two new figures, Supplementary Fig 17-18. The text within the main manuscript has been modified accordingly.

Text added to the main manuscript (Page 13 / Page 14):

'We also studied the influence of vessel flow velocities on microswarm formation. Here, we observed higher occurrence of swarm formation at lower flow rates, we show these results in Supplementary Fig. 16. Given that different blood flows are found in different types of vessels, we further investigated the occurrence of swarm formation and navigation in veins, venules, arteries or arterioles (Supplementary Fig 17, 18). Hence, we mostly observed microbubble formation in veins and venules (with low flows and diameters of 10-40 μm , refer to Fig 4d) compared to arteries and arterioles, see Fig. 4d and Supplementary Fig. 17.'

'We also characterized the occurrence of swarm navigation in veins, venules, arteries or arterioles (Supplementary Fig 17, 18). Here, we observed that microrobot navigation occurred exclusively in veins

and venules (Supplementary Fig 17, 18). Additionally, we confirmed that microbubble swarm formation and navigation was mostly successful inside vessels with diameters between 10 to 40 μm (see Fig 5e).'

Text added to Supplementary Information (Page S27-28):

We have additionally classified each vessel between venules, arterioles, veins and arteries, and we have plotted the frequency of microbubble clustering and navigation for each type of vessel. We have analyzed in total 240 formed clusters inside 25 different blood vessels, see Supplementary Fig 17.

Supplementary Fig. 17. | Study of swarm formation and navigation in arterioles, arteries, venules and veins. The data from these plots has been extracted from a total of 240 formed clusters inside 25 different blood vessels **a**. Schematic representation of a capillary bed **b**. Table shows the diameter range that we considered for each type of vessel, according to cited references and we count the amount of each vessel type that we imaged during experiments. A dominance of veins was imaged as they presented the higher chances for swarm formation and navigation. **c**. Blood flow measurements in different types of vessels. **d**. Number of swarm formation events that were observed in each type of vessel. **e**. Level of clustering in each type of vessel. We calculated the total percentage of flowing bubbles in each vessel that were responsive to acoustics and formed a swarm. The way we performed this calculation is detailed in the section 'Fluorescent microbubble dosage and the level of acoustic response'. **f**. Number of swarm navigation events observed in each different type of vessel.

10. The authors demonstrated successful actuation of microbubbles in the PDMS channel at arbitrary acoustic wave incident angles, which indicates that this acoustic actuation strategy is applicable to complex 3D blood vessels in the brain. However, in the in vivo experiments, it can be seen that multiple microswarms are formed in different parts of the blood vessel within the ultrasound-affected area, why does only one of the microswarms can perform locomotion (Supplementary Video 3)?

Response: We thank the reviewer for the comment. As acknowledged by the reviewer, our in vivo experiments revealed that only a subset of the microbubble swarms exhibited successful navigation, whereas others remained stationary in close proximity to the wall. It became clear that achieving microbubble navigation was more challenging compared to microbubble formation, as it required a sufficient amount of acoustic energy to overcome the drag forces from the blood. Due to these challenges, some microbubble swarms were not able to navigate effectively, which highlights the need for further optimization to improve the efficiency of microbubble navigation in vivo. We have scrutinized the efficacy of the navigation results relative to swarm formation events and we have discussed what parameters are still compromising the optimal navigation of microrobots in vivo. We have added in the Supplementary Information (Page S29) a detailed explanation on the different success rate between swarm formation and swarm navigation, with a new figure Supplementary Fig. 18, where we plot the number of swarm formation events versus the number of swarm navigation events. We have discussed this phenomena also within to the discussion (Page 16):

New text in the discussion (Page 16):

'The actuation system also presents limitations, as seen in Supplementary video 3, where multiple microbubble swarms formed but only one of them was able to navigate effectively. While microswarm formation happened more often, navigation was proven to be more difficult to attain (see Supplementary Fig. 18), especially navigation over long distances (see Supplementary Fig. 11).'

Text added to Supplementary Information (Page S29):

Analysis on the efficacy of microswarm navigation relative to microswarm formation

Microbubble swarm formation and navigation was not achieved with the same rate of efficacy. While swarms formed more easily, they tend to stay at the site of formation, ruled by their secondary Bjerknes interactions to the wall, and navigation was only achieved in certain cases. We have analyzed in each vessel the amount of swarm formation and swarm navigation, see Supplementary Fig 18. It was observed that there is an increasing tendency between swarms forming and their navigation rate, the more swarms formed in a vessel, the more swarms presented navigation. However, if we pay attention to the values for swam formation and navigation, swarm formation ranges between 0 and 25 swarms forming in a vessel, while navigation ranges between 0 and 5 swarms being navigated in a vessel. Note that navigation needs of higher acoustic energies to overcome the blood flow drag forces and to override the secondary Bjerknes adherence of the swarms to the wall, this contributes to the reduced amount of navigation of swarms compared to swarm formation.

Supplementary Fig. 18. | Efficacy of microbubble navigation relative to microbubble formation. Plot showing the amount of microswarms that navigate inside a vessel compared to the amount of swarms that form inside the same vessel.

11. Under the same acoustic signal (490 kHz and 35 VPP), why does the microswarm in Supplementary Video 1 gather in situ while the microswarm in Supplementary Video 3 moves upstream?

Response: We thank the reviewer for the comment. It is important to note that the vessels depicted in Supplementary Video 1 and Supplementary Video 3 are distinct from each other. This implies that the two vessels occupy different spatial locations and have varying relative orientations with respect to the transducer. We have included a new Figure 3 in the manuscript to illustrate how the microbubble navigation behavior is influenced by the specific orientation of the transducer. It is possible for the transducer location to be so misaligned with the target vessel that microbubbles aggregate without displaying any net navigation. We have included an explanation inside the discussion that explains why the same acoustic signal can induce different microrobot responses at different vessels within the brain.

Text from the discussion (Page 15):

‘Additionally, through our experiments in microfluidic device, we showed that the position of an acoustic transducer determines the magnitude and direction of microbubble response. This implies that the transducer positioning impacts the speed and location of swarms self-assembly and movement within a channel. Nevertheless, in all cases microbubbles move along the vessel wall and in the direction of wave propagation, under the constraints of the boundaries of the channel. Based on our findings, we elucidated that a single acoustic signal generated by a transducer can induce a diverse range of microrobot navigation patterns within a three-dimensional network of microchannels. This is particularly significant in scenarios where the relative positions of the vessels to the transducer are variable, and multiple microbubble swarms are influenced by the acoustic signal. To tackle this situation, our software predicted the vessels within the network that would present higher successful swarm formation and navigation. The predictions from the software were successfully validated during in vivo experiments.’

Additionally, we have presented in Supplementary Fig. 18 (Page S29) a greater occurrence of microswarm formation events relative to navigation, which suggests a lower navigational efficiency for microswarms. This highlights the presence of swarm formations that do not engage in navigational behavior.

Supplementary Fig. 18. | Efficacy of microbubble navigation relative to microbubble formation. Plot showing the amount of microswarms that navigate inside a vessel compared to the amount of swarms that form inside the same vessel.

12. Are the acoustic waves that drive microbubbles against flow safe for brain tissue? When need to conduct actuation in deeper regions, the parts close to the transducer are inevitably more affected by the acoustic wave. For example, the measured temperature (Supplementary Figure 5) should be higher in this case.

Response: We thank the reviewer for the comment. Apart from the previous study on temperature increase, we have now performed additional biocompatibility experiments to assess any damage that may be caused by acoustic waves on the mouse brain. We can now demonstrate that the acoustic waves that we used for experiments didn't disrupt or damage neuronal or vascular health. We have added a new section to the Supplementary information (Page S30-32) called 'Biocompatibility studies' with two new Supplementary Fig 21-22.

We have performed new experiments that assess the state of the brain tissue during and after acoustic actuation with 20-30-40VPP and 490 kHz (same settings used previously for microbubble navigation). First, we analyzed the specific effect of acoustic signal alone, without microbubbles injection, on the BBB integrity. Our data showed that no disruption of BBB occurred upon increasing voltages, as no leakage is observed from the blood vessels to the brain tissue (see Supplementary Fig. 20). Furthermore, we combined ultrasound with microbubbles, as gas filled microbubbles are known to amplify the acoustic signal. Microbubbles were injected intravenously, and brain-coupled transducers were activated at 40VPP and 490 kHz. After experiments, mice were sacrificed, and we performed histology studies. Immunohistochemical assessment of the brain tissue after ultrasound actuation revealed neither disruption of the vascular endothelium (Supplemental Figure 21) nor neuronal cell death. Astrocyte activation was minimal and confined to the superficial layer of the cortex. See Supplementary Information (Page S30-32) for full details.

Regarding the effect of temperature when targeting deeper tissues, we have included a new fragment inside the discussion of the manuscript that explains the safety of acoustic waves, including situations where we want to target deeper tissues. We refer to already existing cooling techniques that are being used to apply acoustics in vivo for applications such as focused ultrasound for ablation or ultrasound imaging¹¹⁻¹³.

Text added to discussion (Page 16):

'We use microbubbles with diameters of approximately 1.1-1.4 μm , and a resonance frequency of 490 kHz. Our working frequency is far from the acoustic resonance of the microbubble, reducing detrimental

effects on the surroundings that could arise from behaviors such as microbubble fast strong oscillations or even cavitation. Biosafety and biocompatibility of microbubbles is one important aspect to examine when considering the use of microrobots in vivo, particularly in regard with blood brain barrier damage, neuronal damage and other potential toxic effects. Immunohistochemical assessment of the brain tissue after ultrasound actuation revealed neither disruption of the vascular endothelium (Supplementary Fig 20) nor neuronal cell death (Supplementary Fig 21). Astrocyte activation was minimal and confined to the superficial layer of the cortex (Supplementary Fig 21). Note that using ultrasound there is a potential temperature increase that should be considered in further studies and biocompatibility analyses. When targeting deep regions within the brain, various techniques have already been developed to avoid excessive heating in the tissue, i.e. monitoring the temperature, limiting the exposure time, or the use of cooling techniques¹¹⁻¹³

Within the discussion we have now referred to the Supplementary Information; where we have added the biocompatibility studies. Text added to Supplementary Information (Page S30-32):

Biocompatibility studies

First, we have studied the effect of ultrasounds alone. A mouse C57BL/6 was coupled to two transducers, we applied 20V_{PP} for 3 recordings of 200 seconds each. Then 30V_{PP} for three recordings of 200 seconds each. And finally, 40 V_{PP} for 6 recordings of 200 seconds each. This is the same signal that was used for mouse during navigation experiments. During the recordings we didn't inject microbubbles but we injected Hoechst 33342 Fluorescent Stain that stains cell nuclei and we proved that no leakage was observed during the whole experimental time. Fluorescent signal can be seen inside the vessels but not outside, proving endothelial integrity within the BBB.

Supplementary Fig. 20. | Ultrasound effect on BBB integrity. Hoechst 33342 Fluorescent Stain was injected into the mouse, and increasing voltages were applied to two piezo electric transducers coupled to the mouse head. After 40 minutes of acoustic activation, no leakage or disruption of the BBB was observed, the dye remained inside the blood vessels. Scale bar, 50 μ m.

Gas filled microbubbles that are surrounded by a lipid layer are often used as contrast agents in medicine, thus numerous studies have already proven their biocompatibility. We additionally assessed the biocompatibility of these bubbles for the brain tissue. We have undergone biocompatibility studies to analyze the effect on brain tissue of oscillating microbubbles.

A C57BL/6 mouse was injected with 50 μ L of microbubble solution. After 5 minutes, another 50 μ L were injected into the mouse, same dosage as the one used for microbubble navigation experiments. After one hour of acoustic activation and microbubble navigation in the mouse, the mouse was euthanized and perfused for histology studies. We characterized astrocyte activation via the marker GFAP, endothelial integrity via the marker CD31 and cell viability using DAPI and NeuN, the last one specifically for neurons. The results show that the acoustic activation of microbubbles do not trigger abnormal tissue death. Endothelial cell lining keeps its integrity; however, astrocyte activation is present at the surface of the tissue, see Supplementary Fig. 21.

Supplementary Fig. 21. | Histology immunostaining for biocompatibility analysis. a. Merged image of a brain slice of the mouse after staining with CD31, DAPI, GFPA and NeuN. Scale bar, 50 µm. **b.** Individual images for each cellular maker, CD31, GFPA, NeuN and DAPI, from left to right, top down. The images show endothelial integrity, astrocyte superficial activation, neuronal survival and tissue integrity. **c.** Merged image of a brain slice of the mouse, at a second differentiated spot, after staining with CD31, DAPI, GFPA and NeuN. Scale bar, 50 µm. **d.** Individual images for each cellular maker, CD31, GFPA, NeuN and DAPI, from left to right, top down. The images show endothelial integrity, astrocyte superficial activation, neuronal survival and tissue integrity.

In response to Reviewer 2 comments.

The authors describe an imperfect but highly inventive technique to propel and guide clusters or swarms of microbubbles in brain blood vessels. The work is inventive, and the accomplishment is clear, albeit modest, i.e., a few micrometers of transport per second. Nonetheless, creativity carries the day and I support publication. The expanding use of ultrasound will attract others to this approach.

We greatly appreciate and acknowledge the reviewer's insightful comments. We thank the reviewer for considering our work inventive and creative. We thank also for the constructive feedback, we have now answered and clarified the mentioned queries. The new or modified text has been marked in green within the manuscript and the Supplementary Information. Fragments of this text have also been added to the Reviewer's responses, for more clarity.

1. The authors specify the maximum flow speed that they can move against as 10 mm/s. Is this in the center of the vessel or within a micrometer of the vessel wall?

Response: We thank the reviewer for the question. The flow was measured at the center of the vessel; we mention now this information in the main text (Page 14). For flow measurements, a line scan was performed during acquisition with the two-photon microscope. For clarity we have added to the Supplementary Information (Page S30) a figure that depicts the location within the vessel of the acquired line scan.

Text added to main manuscript (Page 14): 'We measured microrobots upstream velocities up to 1.5 $\mu\text{m/s}$, overcoming flow rates up to 10 mm/s (see Fig 5d). Blood flow was measured at the center of the blood vessels (Supplementary Fig. 19).'

Text added to the Supplementary Information (Page S30):

Measurement of blood flow inside blood vessels in vivo

To image blood flow, we used two-photon microscopy. A line scan was conducted along the center of the imaged vessel, aligned with its direction. By examining this line scan, we were able to determine the velocities of blood flow. This investigation was carried out across blood vessels of varying diameters to establish a correlation between vessel size and the corresponding blood flow. These findings were crucial for the subsequent analysis of microrobots.

Supplementary Fig. 19. | **a.** Image of a vessel imaged with two-photon microscopy showing a line drawn in the middle to perform a line scan and extract blood velocities. Scale bar, 50 μm . **b.** Results from the line scan, the plot shows distance versus time, each black line in the plot represents a flowing particle (i.e. red blood cells).

2. On page 5 the authors state "... upon skull connection the transducer resonance frequency suffered a subtle shift to the left ...". First, I take it that "shift to the left" means to lower frequency. Second, this sounds like the transducer was mechanically loaded, as a loss mechanism leads to a decrease in the peak frequency of a resonator. What is this loading?

Response: We thank the reviewer for the insightful comment. When the transducer is glued to the skull, the total vibrating mass on the transducer is increased, thus, reducing its resonance frequency. The mechanical loading that is being exerted on the transducer comes from the skull itself, which is glued to the transducer. We now mention the effect of mechanical loading within the main text (Page 6) and we reference to the Supplementary Information (Page S5-6) where we explain it in detail.

Text added to the main manuscript (Page 6):

'We conducted an investigation on the transmission and attenuation of ultrasound through an *ex vivo* skull with the aim of manipulating microrobots inside the brain tissue. When a piezoelectric transducer is connected to the skull of a mouse, the effective vibrating mass of the system is increased, which affects both the resonant frequency of the transducer and the intensity of the generated sound wave (see details in Supplementary Information). For this analysis, we bonded a piezoelectric transducer to an *ex vivo* mouse skull and measured the transducer's impedance across a frequency range of 100 kHz to 1 MHz with a step size of 1 kHz using an impedance analyzer. The resulting plot is shown in Supplementary Fig. 3a. We observed a slight leftward shift (~15 kHz) in the resonant frequency of the transducer and an attenuation of the vibration intensity (Supplementary Fig. 3a).'

Explanation added to Supplementary Information (Page S5-6):

Mass loading during transducer-skull coupling

When a mass is added to a piezo transducer, it affects its resonance properties by altering the resonant frequency and the mechanical impedance of the transducer¹⁴.

The resonant frequency of a piezo transducer is determined by its physical dimensions and the elastic properties of the materials used in its construction. When a mass is added to the transducer, it increases the effective mass of the system and changes the resonant frequency. Specifically, the resonant frequency of the transducer decreases as the added mass increases¹⁴.

The mechanical impedance of a piezo transducer is a measure of its resistance to deformation under an applied force. It is directly related to the resonant frequency and the effective mass of the transducer. When a mass is added to the transducer, it increases the effective mass and reduces the mechanical impedance of the system. As a result, the transducer may become less efficient in converting electrical energy into acoustic energy¹⁴.

Gluing a piezo transducer to a surface introduces mass loading to the system and affects its resonance properties, this is exactly what we see in Supplementary Fig 3. During our experiments, we glued a piezoelectric transducer to a mouse skull, thus, increasing the total vibrating mass. The amount of change in resonant frequency will depend on the amount of mass added and the stiffness of the surface¹⁵.

3. Were the drag forces on the microbubbles the same in pial veins as in pial arterioles?

Response: We appreciate the reviewer's question. To facilitate the discussion on drag forces, we will account for the direct correlation between drag forces and blood flow using the formula $F_{\text{drag}} = 6\pi\mu aV$, where F_{drag} represents the drag force, μ is the fluid's viscosity, a is the particle's radius, and V is the fluid's velocity relative to the particle.

We have added new text within the Supplementary Information (Page S26-27) where we explain the relationship between blood flows and drag forces. Where higher blood flows will present higher

impediments to microbubble swarm formation and navigation. Note that pial veins and pial arterioles possess different blood flow velocities; thus, they will exert different drag forces on the microbubbles.

New text in Supplementary Information (Page S26-27):

'To assess the efficacy of microrobot control *in vivo*, we analysed how successful was microrobot formation and navigation in blood vessels, and we studied how the vessel intrinsic characteristic (type, size and blood flow) affects the amount of microrobot control. For all *in vivo* experiments, acoustic signal was kept constant at 490 kHz and 44 V_{PP}.

First, we want to illustrate how higher blood flow values make more difficult microswarm formation. Higher blood flow means higher drag forces that compete with acoustic radiation forces; thus more difficult microbubble aggregations. The calculation of the drag force experienced by a spherical particle in a microscale flow can be performed using the Stokes drag formula, represented by $F_{\text{drag}} = 6\pi\mu aV$. In this equation, F_{drag} denotes the drag force, μ represents the viscosity of the fluid, a stands for the radius of the particle, and V corresponds to the velocity of the fluid relative to the particle. Consequently, blood vessels that are characterized by lower blood velocities, will generate lower drag forces acting upon the manipulated microbubbles. This relationship is illustrated in Fig. 4 of the main manuscript, where the diameter of the blood vessels is demonstrated to be correlated with blood flow velocities.'

Regarding the reviewer's question, pial veins and pial arterioles are known to present different blood flows. **Therefore, it is expected that pial veins and pial arterioles will experience different drag forces.** However, we have further supported this claim experimentally. Per the reviewer's inquiry, our experimental analysis now incorporates an extra characterization, which involves:

- The blood flow velocities present in each vessel type. We investigate the presence of different drag forces in different types of vessels.
- Examining the level of swarm formation and navigation at different blood flow rates. Here we analyze the level of swarm formation at different drag forces.
- The number of swarm formation events, swarm clustering rate and swarm navigation events inside venules, veins, capillaries, arterioles and arteries. We study the impediment on microbubble formation and navigation at different types of vessels.

The drag forces depends on the flow velocity within the vessels. During our experiments we compared blood flow velocity between veins, venules and arterioles. We observed that blood flow in veins is higher than in arterioles, due to their large diameter, see Supplementary Fig 17. However, arterioles in average present higher blood flows than venules^{16,17}. Additionally, when characterizing the efficacy of swarm formation and navigation inside venules or arterioles, we see a dominant formation in venules compared to arterioles, see Supplementary Fig 17.

The higher swarm activation inside venules can be attributed to two features. First, the acoustic damping. Note that arteriole possess much more elastic wall, that adapts to regulate pressure inside blood, while veins and venules lack of wall elasticity; this characteristic makes microbubbles inside veins and venules easier to sense the acoustic wave. On the other hand, the tendency to find lower flows in venules compared to arterioles, also contributes to reduced drag forces inside venules.

The data regarding microbubble formation and navigation in pial veins and pial arterioles is presented within the Supplementary information (Page S28), with two new figures, see Supplementary Fig. 16-17.

Text added to the Supplementary Information (Page S26-28):

We have additionally classified each vessel between venules, arterioles, veins and arteries, and we have plotted the frequency of microbubble clustering and navigation for each type of vessel. We have analyzed in total 240 formed clusters inside 25 different blood vessels, see Supplementary Fig 17.

Supplementary Fig. 17. | Study of swarm formation and navigation in arterioles, arteries, venules and veins. The data from these plots has been extracted from a total of 240 formed clusters inside 25 different blood vessels **a**. Schematic representation of a capillary bed **b**. Table shows the diameter range that we considered for each type of vessel, according to cited references and we count the amount of each vessel type that we imaged during experiments. A dominance of veins was imaged as they presented the higher chances for swarm formation and navigation. **c**. Blood flow measurements in different types of vessel. **d**. Number of swarm formation events that were observed in each type of vessel. **e**. Level of clustering in each type of vessel. We calculated the total percentage of flowing bubbles in each vessel that were responsive to acoustics and formed a swarm. The way we performed this calculation is detailed in the section 'Fluorescent microbubble dosage and the level of acoustic response'. **f**. Number of swarm navigation events observed in each different type of vessel.

4. The size of the microswarm clusters needs to be clarified. Figure 3c shows cluster diameters of up to $30 \mu\text{m}$ while Figures 3d, 3e, 4e, and 4f imply sizes are less than $15 \mu\text{m}$. The text states "... 3 to $10 \mu\text{m}$ in size." What is it?

Response: Note that we have added one full figure to the main text, so previous Figures 3 and 4, are now Figures 4 and 5.

We appreciate the reviewer's feedback and would like to clarify the size range of the microbubble swarms. We agree that the previous variation in numbers could potentially cause confusion. Therefore, we have now modified Figure 4d and we have modified the main manuscript text.

In Figure 3c, we show the smallest and largest swarms that we observed in our experiments, which were $1 \mu\text{m}$ and $40 \mu\text{m}$ respectively. In the new Figure 4d, we plot the size distribution of all swarms between $1 \mu\text{m}$ and $40 \mu\text{m}$. We specifically highlight the majority of swarms, which range from $3 \mu\text{m}$ to

15 μm . For the remaining figures 4e, 5e, and 5f, we focus on further experiments that characterize this majority of swarms ranging from 3 μm to 15 μm .

See new figure 4d:

Fig 4d. The inset at top right illustrates a histogram for the size distribution of all the observed microswarms during experiments (from 1 to 40 μm). The majority of swarms range between 3 to 15 μm , which are plotted in the main histogram plot. These plots show the cluster diameter distribution observed at an acoustic excitation of 490 kHz and 35 V_{pp}. A total sample of 200 clusters was analyzed. Fitting lines have been drawn to summarize the tendency for each plot.

We have additionally modified the main text to clarify this (Page 13):

We then characterized the temporal evolution of swarm size inside blood vessels (Fig 4c). We observed that as hemodynamic forces brought more microbubbles to the activated site, the clusters grew, resulting in swarms of size up to 40 μm until reaching a saturation state (Fig. 4c). However, not all swarms grew at the same rate. The variability in microswarm formation speed, seen in Fig 4c, comes from the complexity of the brain vasculature network and the variable relative position of each blood vessel to the activated transducer. In addition, we studied the size distribution of all microbubble swarms that we observed during experiments, see Fig 4d. We showed that the majority of swarms range was 3-15 μm , with larger swarms found in larger vessels (Fig. 4d). Thus, for the remaining characterization experiments we focused on swarms with these sizes, 3-15 μm .

This text intends to provide additional clarification regarding the microbubble swarm sizes shown in our figures.

5. The authors state "Microswarms continue attracting each other and grow even during navigation." Is it true that vessels are never clogged by the swarms of microbubbles, even small vessels?

Response: We thank the reviewer for the insightful comment. During acoustic activation, microbubbles aggregate to each other and they grow in size. Note that microbubbles are filled with gas, what makes them compressible and able to adapt their shape. Furthermore, the formation of a swarm is a dynamic process in which the swarm can rearrange configuration to adapt to vessel size and pass through capillaries, we show this now also in an additional figure Supplementary Fig. 15. The microbubbles we use are 1.1-1.4 μm , as reported by the provider; thus they are much smaller than blood vessels. Additionally, even if the growth mechanism continues also during navigation, the size doesn't grow indefinitely; the study of growth over time has shown that size of a swarm grows until it reaches a saturation state, which is normally much smaller than the diameter of the containing vessel. All in all, the probability of microbubble swarms to create clogging results to be significantly low, however not completely zero.

The probability to have clogging in the mouse vasculature depends on several factors such as the size of the vessels, the concentration and size of the microbubbles, and the flow conditions in the vessels. Note that during experiments, we sometimes observed incipient clogging situations. In these cases, we switched off the acoustic signal and the microbubbles were washed away, so we could recover normal flow in the vessel after some seconds. We have added this topic to the section 'In vivo self-assembly of microrobots in mouse brain vasculature' (Page 13) of the main manuscript; where we discuss the observation of incipient clogging in the mouse vessels and we additionally refer to a new section within the Supplementary Information (Page S25-26), where we explain in detail the factors that contribute to incipient clogging and where do we find them mostly. We explain as well that switching off the acoustics has been seen to enable the recovery of normal flow within the affected vessels. See also new Supplementary Fig. 17b.

In the end, it is still essential to carefully control the concentration and size of microbubbles used in medical applications, and consider the flow conditions and vessel size when using them in diagnostic or therapeutic procedures. Additionally, thorough testing and monitoring should be performed to ensure that microbubbles are used safely and effectively in medical applications.

Text added to main manuscript (Page 13):

It is still essential to carefully control microbubbles swarms when used in medical applications. During our experiments, we observed incipient cases of vessel clogging in approximately 3% of the analyzed vessels, primarily in small capillaries and venules. These cases were identified by a sudden reduction in the flow velocity of the affected vessel and the presence of an accumulation of microbubbles (Supplementary Fig. 15). In all these cases, the vascular patency and blood flow were completely recovered when acoustic was turned off.

New text in Supplementary Information (Page S25-26):

Microbubbles used for the experiments are much smaller than the vessel diameters, however, upon acoustic activation, the formation of swarms can result into sizes that come close to the diameter of the blood vessels, and thus clogging becomes an important factor to consider.

We experienced the appearance of incipient clogging events; these were identified by a sudden reduction in the flow velocity of a vessel and the presence of an accumulation of microbubbles. We characterized the number of times that we saw these events during in vivo experiments, compared to the total number of vessels analyzed, and we calculated a total of 3% of vessels that presented this situation. Importantly, after turning off the acoustics, the accumulation of microbubbles was cleared from the vessel, see Supplementary Fig. 15, and normal blood flow was recovered after some time.

We characterized the size of the vessels where this situations happened; we observed a dominance of capillaries with a size range between 1 and 10 μm in diameter, see Supplementary Fig. 15.

Supplementary Fig. 15. | Cases of incipient vessel clogging. a. Plot that shows the size distribution of those vessels that presented incipient vessel clogging. We can see a dominance of small vessels

like capillaries and venules. **b.** Microscope image taken with the two-photon microscope during a clogging event. Vessels shown are venules. After turning off the acoustics, microbubbles were washed away and normal flow was recovered. Scale bar, 50 μm .

6. The authors need to state if the vessels in Figures 4a and 4c are venules or arterioles.

Response: We thank the reviewer for the question. Previous Figure 4 is now Figure 5. The mentioned vessels are venules and we have now added this information to the figure caption:

Fig. 5 | Microswarm upstream navigation in mouse brain vasculature. **a.** At the top, the schematic illustrates the first possible scenario for microswarm navigation: downstream navigation accompanied by simultaneous growth. **In the images below, we observe a microswarm (highlighted in green through manual overlay) inside a venule.** Over time the microswarm successfully navigates along a controlled downstream trajectory, resisting the influence of the background blood flow. Eventually, the microswarm is seen to be attracted to a second microswarm that was already positioned near the wall. The acoustic signal was 490 kHz and 35 V_{PP} . Scale bar is 30 μm . **b.** At the top, the schematic depicts the second possible scenario of microswarm navigation: upstream movement without growth. **In the images below, we observe the movement of a microswarm (indicated by a green overlay) inside a venule.** The microswarm is manually tracked as it navigates along a vessel wall, counter to the direction of blood flow. The acoustic signal was 490 kHz and 44 V_{PP} . Scale bar is 30 μm . **c.** At the top, the schematic illustrates the final possible scenario: microswarm navigation upstream accompanied by simultaneous growth. **In the images below, we observe the movement of a microswarm (indicated by a green overlay) inside a venule.** The microswarm is manually tracked as it progresses against the flow, gathering more bubbles along its path (Supplementary Video 3). The acoustic signal was 490 kHz and 35 V_{PP} . Scale bar is 30 μm . **d.** The plot displays the blood flow velocities typically observed in the mouse vasculature, specifically for vessel diameters where microswarm formation and navigation were observed. We have drawn a linear fit line with $R^2=0.7839$. **e.** The plot illustrates the relationship between the observed upstream velocity of microswarms and both the vessel diameter and swarm diameter. The acoustic signal was 490 kHz and 35 V_{PP} . Each point is the average of five measurements and error bars are the standard deviation. **f.** The correlation between vessel size and cluster diameter demonstrates a tendency for larger swarms to form in larger vessels. Each point is the average of five measurements and error bars are the standard deviation. R^2 for the fitting line is 0.4719. The acoustic signal was 490 kHz and 35 V_{PP} .

References:

1. Fonseca, A. D. C., Kohler, T. & Ahmed, D. Ultrasound-Controlled Swarmbots Under Physiological Flow Conditions. *Adv. Mater. Interfaces* **9**, 2200877 (2022).
2. Fonseca, A. D. C., Kohler, T. & Ahmed, D. Navigation of Ultrasound-controlled Swarmbots under Physiological Flow Conditions. 2022.02.11.480088 Preprint at <https://doi.org/10.1101/2022.02.11.480088> (2022).
3. Chugh, B. P. *et al.* Measurement of cerebral blood volume in mouse brain regions using micro-computed tomography. *NeuroImage* **47**, 1312–1318 (2009).
4. Piscaglia, F. & Bolondi, L. The safety of Sonovue® in abdominal applications: Retrospective analysis of 23188 investigations. *Ultrasound Med. Biol.* **32**, 1369–1375 (2006).
5. Laugesen, N. G., Nolsoe, C. P. & Rosenberg, J. Clinical Applications of Contrast-Enhanced Ultrasound in the Pediatric Work-Up of Focal Liver Lesions and Blunt Abdominal Trauma: A Systematic Review. *Ultrasound Int. Open* **3**, E2–E7 (2017).
6. Calisan, M., Talu, M. F., Pimenov, D. Y. & Giasin, K. Skull Thickness Calculation Using Thermal Analysis and Finite Elements. *Appl. Sci.* **11**, 10483 (2021).
7. Photinos, P. Propagation of sound waves. in *The Physics of Sound Waves (Second Edition): Music, instruments, and sound equipment* (IOP Publishing, 2021). doi:10.1088/978-0-7503-3539-3ch2.
8. Embleton, T. F. W. Mean Force on a Sphere in a Spherical Sound Field. I. (Theoretical). *J. Acoust. Soc. Am.* **26**, 40–45 (2005).
9. Crocker, M. J. *Handbook of Acoustics*. (John Wiley & Sons, 1998).
10. Dayton, P., Klibanov, A., Brandenburger, G. & Ferrara, K. Acoustic radiation force in vivo: a mechanism to assist targeting of microbubbles. *Ultrasound Med. Biol.* **25**, 1195–1201 (1999).
11. Deardorff, D. L., Diederich, C. J. & Nau, W. H. Air-cooling of direct-coupled ultrasound applicators for interstitial hyperthermia and thermal coagulation. *Med. Phys.* **25**, 2400–2409 (1998).
12. Diederich, C. J. Ultrasound applicators with integrated catheter-cooling for interstitial hyperthermia: Theory and preliminary experiments. *Int. J. Hyperthermia* **12**, 279–297 (1996).
13. Francica, G. & Marone, G. Ultrasound-guided percutaneous treatment of hepatocellular carcinoma by radiofrequency hyperthermia with a 'cooled-tip needle'. A preliminary clinical experience. *Eur. J. Ultrasound* **9**, 145–153 (1999).
14. Skeie, H. Electrical and Mechanical Loading of a Piezoelectric Surface Supporting Surface Waves. *J. Acoust. Soc. Am.* **48**, 1098–1109 (1970).
15. Ramakrishnan, N., Nemade, H. B. & Palathinkal, R. P. Resonant Frequency Characteristics of a SAW Device Attached to Resonating Micropillars. *Sensors* **12**, 3789–3797 (2012).
16. Wang, L. *et al.* Vessel sampling and blood flow velocity distribution with vessel diameter for characterizing the human bulbar conjunctival microvasculature. *Eye Contact Lens* **42**, 135–140 (2016).
17. Qi, Y. & Roper, M. Control of low flow regions in the cortical vasculature determines optimal arterio-venous ratios. *Proc. Natl. Acad. Sci.* **118**, e2021840118 (2021).

REVIEWER COMMENTS

Reviewer #1 (Remarks to the Author):

In the resubmitted manuscript, the authors have addressed most of my previous questions in detail, and corresponding revisions have been made to significantly improve the quality of the manuscript. Nevertheless, some of my concerns remain.

The authors propose a method for navigating microbubbles in branched and curved structures using combinatorial transducer actuation and validate it in PDMS devices. However, there may be some problems in practical applications, such as whether there is enough suitable space to place transducers and whether some transducers that are far away from the microbubble can provide efficient actuation. Therefore, it is suggested to verify the applicability of this method by performing controlled navigation of microbubbles in practical 3D branched blood vessels.

In addition, the problem of generating multiple microswarms in areas affected by acoustic waves persists. The software created by the authors to predict the behavior of microbubbles can help coordinate multiple microswarms to avoid adverse effects. The authors conduct in vivo experiments to verify that microswarms are generated at the predicted locations, and a further question is why the predicted moving direction and velocity are not validated based on these microswarms.

Reviewer #2 (Remarks to the Author):

I am pleased with the completeness of the author's response and their clarifications to the text and additions to the manuscript. I look forward to publication of the manuscript.

In response to Reviewer 1 comments.

In the resubmitted manuscript, the authors have addressed most of my previous questions in detail, and corresponding revisions have been made to significantly improve the quality of the manuscript. Nevertheless, some of my concerns remain. The detailed comments are as follows.

Response: We greatly appreciate and acknowledge the reviewer's insightful comments. We also thank the reviewer for considering that the revised version of the manuscript has significantly improved its quality. We have addressed the remaining concerns by performing the required validation experiments. We have performed additional *in vivo* experiments to demonstrate the feasibility of microbubble manipulation through branches also in practical vessels. See new Supplementary Video S4. Additionally, we have incorporated *in vivo* validation of the MATLAB software predictions concerning the trajectory and velocity of microbubbles upon transducer activation. Finally, we delve into strategies for minimizing the impact of un-focused microswarm formation on the therapeutic efficacy of microrobots.

We believe that these additions have not only enhanced the overall quality of the paper but also provided valuable clarity to the questions posed. The new or modified text has been marked in blue within the manuscript and the Supplementary Information. Fragments of this text has also been added to the Reviewer's responses for more clarity. The figures labeled as 'Figure R_' are exclusively intended for the reviewer's response.

The authors propose a method for navigating microbubbles in branched and curved structures using combinatorial transducer actuation and validate it in PDMS devices. However, there may be some problems in practical applications, such as whether there is enough suitable space to place transducers and whether some transducers that are far away from the microbubble can provide efficient actuation. Therefore, it is suggested to verify the applicability of this method by performing controlled navigation of microbubbles in practical 3D branched blood vessels.

Response: We thank the reviewer for the insightful comment. As per the reviewer's suggestion, we have conducted an additional *in vivo* experiment involving microrobot manipulation within the mouse brain branched vessels. We have incorporated explanations regarding the setup and technical intricacies employed for the experiments. Additionally, we have added one new **Supplementary Figure**, one **Figure** to the manuscript and a new **Supplementary Video** to show the implementation of microrobot manipulation within the actual 3D branched blood vessels of the mouse brain.

Firstly, the reviewer expressed concern regarding the space that is available in the mouse head for the implementation of a combinatorial transducer manipulation approach. Thus, we now discuss in the Supplementary Information (Page S31) the number of transducers that the mouse head permits to incorporate enabling efficient microbubble actuation through branches. See Figure R1.

Figure R1. Placement of piezo transducers around the cranial window on the mouse head. Photo taken from the mouse head after placement of the 4 piezo transducers. These transducers were used for microrobot manipulation inside the brain of the mouse.

For the purpose of validating the manipulation in mice, we previously opted for an actuation setup comprising two transducers. Responding to the reviewer's suggestion, we have now incorporated four transducers on the mouse head. This modification enables us to achieve a broader range of directions during microrobot navigation *in vivo*. Importantly, the average size of the mouse head we used was 1.5 mm in surface diameter, while the cranial window and the transducer's size was consistently 3 x 3 mm. Hence, incorporating more than four transducers presented difficulties. Nevertheless, with four transducers, we still have sufficient degrees of freedom to demonstrate the steering principle for maneuvering microrobots using combinatorial transducer activation. Please refer to new Figure 7 and the new Supplementary Video for visual representation. We also acknowledge that decreasing the number of transducers in the system leads to a reduction in the precision with which we can direct acoustic waves in a desired direction. We now mention this phenomena also within the discussion of the main manuscript (Page 15).

Importantly, there are alternative viable approaches to achieve combinatorial actuation of microrobots *in vivo*, using a large number of transducers within the system. One option is to employ higher animal models. By using bigger animal models, there is more space available, allowing for the incorporation of a greater number of transducers. Alternatively, the size of the transducers can be altered. Presently, transducers come in various sizes, and by miniaturizing them to a micro-scale we could also implement a higher number of transducers on the mouse head.

Text added to the discussion of the main manuscript (Page 16):

We used 2P microscopy for the real time imaging of microrobots, which limited the study to be performed at superficial tissue layers. For the acoustic implementation, we used a maximum of four transducers, due to the constrained space on the mouse head surface area. While four transducers are adequate to showcase the steering principle for maneuvering microrobots through branches, they do result in reduced precision for achieving efficient wave directionality required for microrobot navigation.

New text in the Supplementary Information (Page S30):

Combinatorial transducer activation implemented on the mouse head

Importantly, in our previous microfluidic experiments, we utilized an actuation set-up involving up to 18 transducers. However, for the purpose of validating the manipulation in mice, we opted for a simplified setup comprising only 4 transducers. The primary reason for this simplification was the challenge of effectively coupling the transducers to the mouse skull. The transducers we employed had dimensions of 3 x 3 mm, whereas the average size of the mouse head we used was 1.5 mm. Consequently, the available space within the mouse for accommodating the transducers is constrained. Nevertheless, 4 transducers give us enough degrees of freedom to show the steering mechanism of microrobots through branches, see Figure 7. The transducers were linked to a system of switches and interconnected to a single function generator. This setup enables each transducer to be activated independently, while all of them receive the same acoustic signal. For more intricate transducer actuation setups, employing higher animal models could provide additional space for optimal coupling of the transducers.

The reviewer also poses a question regarding whether some transducers that are far away from the microbubble can provide efficient actuation. To address this concern, while navigating through branched vessels, we selected locations situated at a significant distance from the activated transducer. In this experiment the cranial window has a length of 3 mm and we imaged microbubble navigation at ~2.7 mm from the activated transducer. See Supplementary Fig 18. The findings from this investigation have been incorporated into a new section of the Supplementary Information (Page S33), and into a new Supplementary Figure 18.

With this test we assessed the navigation capabilities within our constrained field of view, defined by the cranial window. However, it is important to note that our field of view is restricted by the size of the cranial window, and therefore, we are currently unable to validate the technology's capabilities for

working at distances beyond this limitation (throughout the whole brain hemisphere). Nonetheless, for the clinical use of this technology, a local manipulation approach offers significant benefits. Many diseases, such as tumors, are typically localized at specific region in space. As a result, our manipulation setup can be positioned near the lesion site in each case, facilitating precise manipulation within that specific region. This approach holds great promise for targeted and effective treatments. For future work at regions deep inside the brain, we have already provided *ex vivo* measurements of the acoustic pressure below a mouse skull, as shown in Supplementary Fig 3.

New text to the Supplementary Information (Page S30-31):

Manipulation of microrobots *in vivo* at far distances from the transducer.

We successfully validated the manipulation capabilities of microrobots at regions within the cranial window that are far from the activated transducer. To achieve this, we conducted imaging at the lower end of the cranial window (refer to the red square in Supplementary Fig 18a) while activating Piezo transducer 2, positioned at the upper end of the cranial window (~2.7 mm distance between transducer and ROI 2). Through this experiment, we demonstrate that a single transducer can generate sufficient acoustic pressure to move microrobots with working distance up to 2.7 mm in any region of the window. The results revealed microbubbles forming swarms and navigating in the direction of wave propagation when the transducer was activated at 45 V_{PP} and 490 kHz.

Supplementary Fig 18. Validation of microrobot formation and navigation at distances far from the activated piezo transducer. **a.** Microscope image of the cranial window vasculature network. We used 4 transducers for microbubble navigation inside this network and we show their location relative to the acquired image. **b.** We show the formation of two different microbubble swarms labelled as microswarm 1 and 2. Microbubbles in microswarm 1 are manipulated down, by piezo transducer 2. Microbubbles in microswarm 2 self-assemble at the wall. No navigation was seen for this group of microbubbles. After image processing (see Materials and Methods), the microswarms have been colored in green while non-responding single microbubbles that flow downstream have been colored in red. This image results from a stack of 2 different frames taken between 0 and 10.1 seconds of video recording. Scale bar is 20 μm .

Finally, we performed manipulation of microrobots in blood vessel branches. We show the movement of microrobots from one vessel into another, showcasing their application inside branched 3D blood vessels. Importantly, the navigation of these microrobots remains relatively slow, posing challenges in demonstrating intricate manipulation paths *in vivo*. Nonetheless, we present initial results that showcase the potential applications of these microrobots. These results provide additional support for the results presented in the previous version of the revised manuscript, where we demonstrated microrobot manipulation through 3D branched vasculature networks in microfluidic setups. We have included new text to the main manuscript section 'Navigation of microrobots inside the mouse brain vasculature', a

new figure to the main manuscript 'Figure 7', and a **new Supplementary Video S4**. New Supplementary Video caption:

Supplementary Video 4 (separate file). Microswarm navigation in 3D branched vessels (40.7 MB)

The video corresponds to Fig. 7C. This video shows navigation of a swarm in a branched vessel trajectory. An inset on the left shows the vasculature within the cranial window and the relative position of the transducers. Four transducers were used for microbubble navigation through complex trajectories. On the right, we see the video of the navigation event. First, transducer 2 is activated, moving the swarm downwards, against the flow. Subsequently, transducer 2 is turned OFF and transducer 1 is turned ON, moving the microswarm into a smaller vessel to the right. The swarm moves to the right along the second vessel.

New text to the main manuscript (Page 15):

Ultimately, we conducted experiments demonstrating the precise control of microswarm movement along designated pathways within the intricate vascular network, encompassing branched vessels within the living brain. As demonstrated in our in vitro experiments, microbubbles could be effectively guided through curved and branched trajectories using combinatorial actuation of transducers. Despite facing challenges such as higher drag forces and dampened acoustic waves, we successfully implemented this technology in vivo. To achieve this, we strategically positioned four transducers, labeled from 1 to 4, around the mouse's head, as illustrated in Figure 7. As shown in Figure 7b, we first activated Transducer 3 and the microswarm moved to the left. Subsequently, we activated Transducer 2 (deactivate Transducer 3) and microbubbles move downwards, towards a second and smaller vessel. By skillfully combining different transducers actuation, we successfully moved microbubbles from one vessel to another, as depicted in Figure 7b-d and Supplementary Video 4. These findings highlight the promising capabilities of our approach for navigating microbubbles through complex branched networks in the living brain.

Fig. 7 | Microswarm navigation through branched vessels in vivo in the mouse brain using combinatorial ultrasound actuation. a. Microscope image of the cranial window vasculature network. We used 4 transducers for microbubble navigation inside this network and we show their location relative to the acquired image. We also show the spatial locations where we performed microswarm navigation, named as 'ROI 1 and ROI 2'. **b.** Microbubbles are manipulated first left, by piezo transducer 3 and then down by piezo 2 transducer 2. The

microswarm is manually tracked as it navigates along a vessel wall, counter to the direction of blood flow. The acoustic signal was 490 kHz and 45 V_{PP}. This image results from a stack of 4 different frames taken between 0 and 6.7 seconds of video recording. Scale bar is 20 μm. **c.** Microbubbles are manipulated first down, by piezo transducer 2 and then right by piezo transducer 1. The microswarm is manually tracked as it navigates along a vessel wall, counter to the direction of blood flow. The acoustic signal was 490 kHz and 45 V_{PP}. This image results from a stack of 4 different frames taken between 0 and 50.1 seconds of video recording. Scale bar is 20 μm.

In addition, the problem of generating multiple microswarms in areas affected by acoustic waves persists. The software created by the authors to predict the behaviour of microbubbles can help coordinate multiple microswarms to avoid adverse effects. The authors conduct *in vivo* experiments to verify that microswarms are generated at the predicted locations, and a further question is why the predicted moving direction and velocity are not validated based on these microswarms.

Response: We appreciate the reviewer's comment. To clarify, the predicted moving direction and velocity of microswarms have now been described in detail. We have additionally validated these results *in vivo*. We show these results in a **new Figure 4** and three new **Supplementary Fig:** 14, 15 and 17.

In the previous revised version of the manuscript, we discussed the formation of swarms in the background, that happens as consequence of a non-focused ultrasound signal and that results in swarms forming and moving also along the vessels that we don't focus on. We developed a MATLAB software that enabled us to predict microbubble behavior inside the vasculature network and thus, mitigate uncertain effects. Still, generating multiple microswarms in the areas affected by acoustic waves is an intrinsic consequence of the technique presented in this study. This issue persists in all field-driven manipulation systems, such as for example those relying on magnetic fields. Additionally, from a biological point of view, the brain did not show any deleterious or damaging effect on the brain cells (analysis of the whole brain was done by immunohistochemistry). Even if multiple microswarms were generated in different areas, these microswarms degenerate directly when the acoustic wave is turned off and they are washed away, as shown along the manuscript. We now additionally mention within the discussion the existence of techniques that confine the acoustic field to specific regions in space^{1,2}. These techniques could potentially be applied to the brain and be combined with our developed microrobotic manipulation approach to mitigate the aforementioned phenomena.

New text added to the discussion (Page 16-17):

While the initial results in planned navigation of microbubble-based microrobots are promising, there are factors that require further optimization and control. **Generating multiple microswarms in the areas affected by acoustic waves is an intrinsic consequence of the technique presented in this study. Eliminating the phenomenon entirely is not currently feasible. Thus, our software-based prediction approach focuses on acknowledging and minimizing its impact to the greatest extent achievable.** Our preliminary results show that it is possible to predict microbubble movement along a defined region of the brain vasculature, as outlined in the Supplementary Info; however, we can still not control microswarms in multiple vessels simultaneously. **There are currently techniques that tune the shape of the acoustic pressure to specific regions^{1,2}. Integrating these techniques with our manipulation technology, particularly in the context of the brain, would yield significant benefits.**

Based on the Reviewer's suggestion, we have added directionality and speed to our software. We now include visual representations of microrobot direction upon each transducer activation, which were predicted using MATLAB software (Supplementary Fig 14). Additionally, we have conducted *in vivo* experiments to validate these predictions (Supplementary Fig 17).

In our earlier software analysis of microbubble velocity, we only considered the influence of blood vessel orientation. However, velocities are not only affected by vessel orientation but also by the distance from the transducer. To address this, we have introduced a visual representation that demonstrates how microrobot velocities change with varying distances from the transducer (Supplementary Fig 15).

Moreover, we explain how this information should be effectively integrated with the data concerning vessel orientation. We have further validated these predictions through *in vivo* experiments (Figure 4).

The inclusion of these new predictions provides more accurate and precise information on microrobot movement upon a specific transducer activation, enabling us to design transducer activation strategies that effectively minimize the uncertainty of these "background" microswarms. We have added one new Figure 4 and three new figures to the Supplementary Information, Supplementary Figures 14, 15 and 17, and new supplementary text to explain the results from the new predictions performed by the software. To set the context on the previous state of our results, we refer to Supplementary Fig 13.

New text added to the Supplementary Information (Page S21-23):

Once we have stored data on the spatial relationship between the activated transducer and the blood vessels, we used this data to anticipate the path that microrobots would traverse within each vessel following activation of a designated transducer. Upon activating each transducer, microrobots tend to move in the direction of wave propagation. This behavior predominantly occurs in vessels that are appropriately aligned with the wave propagation direction. To visually convey this information, we generated images of the reconstructed vasculature network, where different colors indicate different vectorial directions of the microrobot trajectory, see Supplementary Fig 14. For example, if a transducer is located at 180° (see angle reference in Supplementary Fig 14a), the microbubbles will move to the (1,0) vectorial direction. Thus in this case, blood vessels where microbubbles are present, will be colored in cyan blue. Within these images, the intensity of the color serves as an indicator of the effective micro vessel orientation. Brighter vessels correspond to those in which microrobots exhibit increased movement and higher velocity. For our example, those vessels located horizontally in the image will show bright blue colors, while vertical vessels will show the darker tones of blue (see Supplementary Fig 14c)

Within the vessels that exhibit a more favorable orientation in relation to the transducer, there will also be relative changes in velocity. These changes are attributed to the varying distances between the vessels and the transducer. Thus, we additionally depicted the velocity change that occurs between different blood vessels due to their relative distance to the activated transducer. As explained in Supplementary Fig 6, microbubble velocity decreases as their distance from the transducer increases. We acknowledge that acoustic pressure diminishes with increasing distance from the transducer. Furthermore, we recognize that acoustic pressure directly correlates with the velocity of microbubbles, as depicted in Supplementary Figure 6. Leveraging the knowledge of the distance between the transducer and each blood vessel, we generated a reconstructed image of the vasculature network where brighter colors indicate shorter distances to the transducer, and as consequence higher microbubble velocities, see Supplementary Fig 15. It is noteworthy to mention that our experimental measurements, as illustrated in Supplementary Figure 3, include acoustic pressure readings beneath a mouse skull at various depths and distances, we also include this information in Supplementary Fig 15. The maximum distance from the transducer within our field of view is half a millimeter. Notably, we observed a small decay in experimental acoustic pressure within this distance range. Ultimately, by combining the information from Supplementary Fig 13, 14, 15; we can derive the vessels where microrobot formation will take place, their velocity and their movement direction.

On our study we have applied this approach to the vasculature maps of mice and reconstructed the skeleton structure of the vasculature map using a color map, where vessels appearing brighter indicate a higher degree of impact from the actuated piezo. The experiments *in vivo* proved that the identified brighter vessels showed microswarm formation events, while the darker vessels didn't, see Supplementary Fig 16. In addition to the aforementioned validations, we have further confirmed that the formed microswarms exhibit movement in the direction of wave propagation. This validation is supported by the findings presented in Supplementary

Figure 17, which provide visual evidence of the directional movement of the microswarms aligned with the wave propagation. Furthermore, we have also validated the velocity and effectiveness of microswarm movement when the microswarms are positioned in close proximity to the transducer. Figure 4 demonstrates the improved movement efficiency of the microswarms in this scenario.

Supplementary Fig. 14. Prediction of microrobot movement direction based on transducer position. a. Schematic of the colormap used to describe microrobot movement directionality. The direction of microrobots has been described by vector units which have their (0,0) at the center of the acquired image. Each vector unit has been assigned to a color, leading to a polar-coordinate colormap system. The black dots on the graph indicate the locations of the transducers employed for images b and c. When referring to the position of the transducer angle, the line connecting (0,0) to (1,0) signifies an angle of 0°. Our measurements of angles follow a counterclockwise direction. Thus, the two black dots on the graph correspond to angles of 60° and 180°. **b,c.** Reconstruction of the vasculature network where the color of the vessels represents the direction where microrobots would move upon transducer activation. Different intensities (light or darker color) has been used to represent those vessels where movement is expected to occur more efficiently. Transducers at 60° and 180° have been used respectively.

Supplementary Fig. 15. Prediction on microrobot velocity changes due to vessel distance from the transducer. **a.** Schematic showing the position of transducers. To encompass the entire field of view, the maximum distance of the visualization window (measured diagonally) has been selected as the diameter of the circle within which the transducers are positioned, 500 μm . **b.** In this plot we present the decay of acoustic pressure versus distance, specifically focusing on the range of 500 μm . This range represents the pressure values within our field of view. The relationship between acoustic pressure and microbubble velocities is depicted in an equation above the plot. Additionally, we provide a colormap representation below the plot, indicating the color assigned to each distance value. This colormap specifically corresponds to the color coding used in 'c'. **c.** Reconstruction of the vasculature network for the activation of three different transducers. In each case, the color of the vessels represents their distance to the respective activated transducer. This color scheme allows for a visual representation of the varying distances between the transducers and the vasculature network.

Supplementary Fig. 17. *In vivo* validation of microrobot trajectory predictions. In this figure left images coincide with front transducer activation, while right images coincide with side transducer activation. See positions of front and side transducers in Supplementary Fig. 16. **a.** Reconstruction of vasculature lattice. The color assigned to each vessel indicates the trajectory that the microrobots will follow as they move within these vessels. To assist in interpreting the colormap, a reference is provided on the right side of the visualization (see also Supplementary Fig. 14). The color intensity (darker or brighter color) indicates those that will exhibit enhanced microswarm navigation when activated by the transducers. This enhanced navigation is attributed to the relative orientation of these vessels with respect to the activated transducer. **b.** Two-photon images from the mouse brain vasculature upon activation of front and side transducer respectively. During activation of front transducer, the microswarms are not present within vessels where navigation is efficient, so no movement is observed (see Figure 4 for further explanations). During activation of side transducer, the microswarms show movement in the predicted direction.

Text added and modified in the main manuscript (Page 12):

We then developed a MATLAB code to store in vectors the relative angles and distances of each vessel with respect to the activated transducers and we reconstructed a color-coded map of the vasculature lattice, see Fig 4 and Supplementary Fig 13. The developed software identified vessels that were located orthogonally to the activated transducer (i.e ultrasound

wave coming at 0°) where microbubbles were expected to exhibit more efficient swarm formation and navigation (see Fig 4a and Supplementary Fig. 12-15). Subsequently, the system generated a visual representation of the direction in which microrobots move upon activating each transducer, see Supplementary Fig 17, and the distance of each vessel to the activated transducer, see Fig 4b. We finally validated the predictive data *in vivo* in pial vessels (see Fig 4c,d and Supplementary Fig 16-17). Specifically, we focused on the activation of the so-called ‘side transducer’, located on the lateral side of the cranial window and the ‘front transducer’, located on the upper side of the cranial window. We analyzed the movement of microbubbles in two differentiated vessels, labelled as Vessel 1 and 2, see Supplementary Information. The resulting microbubble velocities are shown in Fig 4d. Our results show a strong correlation between the *in silico* and *in vivo* data allowing to predict precisely the vessels where microswarm formation and navigation occur (Fig 4d and Supplementary Fig 14 b,c).

Fig. 4 | Prediction of microrobot behavior inside a vasculature network and validation *in vivo*. Left images coincide with front transducer activation, while right images coincide with side transducer activation. See positions of front and side transducers on the mouse, in Supplementary Fig 16. **a.** Reconstruction of vasculature lattice. The colormap indicates those vessels that will show higher or lower speeds of microswarm formation and navigation due to their relative orientation with the activated transducer. Angle of 0° indicates that the line connecting the image center to the transducer is parallel to the line corresponding to the vessel length. Angle 90° indicates that these two lines are perpendicular. **b.** Reconstruction of vasculature lattice. The color assigned to each vessel indicates the distance of the blood vessel to the activated piezo transducer. **c.** Two-photon images from the mouse brain vasculature upon activation of front and side transducer respectively. Each image corresponds to a stack of all the

frames recorded during the indicated time duration. After image processing (see Materials and Methods), the microswarms have been colored in green while non-responding single microbubbles that flow downstream have been colored in red. Scale bar is 50 μm . **c.** Recorded microswarm downstream, velocities in two different vessels of the vasculature network, named as Vessel 1 and Vessel 2. See positions of Vessel 1 and 2 in 'c'. At time 0, no acoustic signal is being applied. Recorded velocities correspond to natural downstream velocity. At t_1 , the front transducer is activated at 45V_{PP} and 490 kHz. At t_2 , the front transducer is deactivated and the side transducer is turned on at 45V_{PP} and 490 kHz.

References:

1. Melde, K., Mark, A. G., Qiu, T. & Fischer, P. Holograms for acoustics. *Nature* **537**, 518–522 (2016).
2. Melde, K. *et al.* Acoustic Fabrication via the Assembly and Fusion of Particles. *Adv. Mater.* **30**, 1704507 (2018).

REVIEWERS' COMMENTS

Reviewer #1 (Remarks to the Author):

The authors conduct additional experiments to demonstrate the promising potential of the proposed ultrasound-activated microrobots. All my questions have been addressed, and the revised manuscript is of good quality to be published in Nature Communications.

In addition, there are two descriptions that need to be checked:

Page 15 of the manuscript: "...as depicted in Figure 7b-d and Supplementary Video 4." Please check if the figure label is correct.

Page 30 of the Supporting Information: "...the mouse head we used was 1.5 mm." Please check if the size of the mouse head given is correct.

Point by point response to Reviewer 1 comments

The authors conduct additional experiments to demonstrate the promising potential of the proposed ultrasound-activated microrobots. All my questions have been addressed, and the revised manuscript is of good quality to be published in Nature Communications.

Response: We greatly appreciate and acknowledge the reviewer's insightful comments. We also thank the reviewer for considering that the manuscript is of good quality and that the additional experiments demonstrate the promising potential of the proposed ultrasound-activated microrobots.

In the revised version of the manuscript we have addressed the last corrections highlighted by the reviewer.

In addition, there are two descriptions that need to be checked:
Page 15 of the manuscript: "...as depicted in Figure 7b-d and Supplementary Video 4." Please check if the figure label is correct.
Page 30 of the Supporting Information: "...the mouse head we used was 1.5 mm." Please check if the size of the mouse head given is correct.

Response: We thank the reviewer for these comments. In Figure 15 of the manuscript, we have updated the label to 'Figure b,c'.

Additionally, we corrected the units for the measurements taken on the mouse head (this text has now been moved to Methods):

Text in Methods, Page 19:

The transducers we employed had dimensions of 3 x 3 mm, whereas the average width of the mouse head we used was 1.5 cm and length 2 cm.